# Near-ice Hydrographic Data from Seaglider Missions in the Western Greenland Sea in Summer 2014 and 2015

Katrin Latarius[1,2], Ursula Schauer[2], Andreas Wisotzki[2]

[1]Bundesamt für Seeschifffahrt und Hydrographie (Federal Marine and Hydrographic Agency), 20359 Hamburg, Germany
[2]Alfred Wegener Institute, Helmholtz Centre for Polar and Marine Research, 27570 Bremerhaven, Germany

*Correspondence to:* Katrin Latarius (katrin.latarius@bsh.de)

**Abstract.** During summer 2014 and summer 2015 two autonomous Seagliders were operated over several months close to the ice edge of the East Greenland Current to capture the near-surface freshwater distribution in the western Greenland Sea. The mission 2015 included an excursion onto the East Greenland Shelf into the Norske Trough. Temperature, salinity and drift data were obtained in the upper 500 to 1000 m with high spatial resolution.

The data set presented here gives the opportunity to analyse the freshwater distribution and possible sources for two different summer situations. During summer 2014 the ice retreat at the rim of the Greenland Sea Gyre was only marginal. The Seagliders were never able to reach the shelf break nor regions where the ice just melted. During summer 2015 the ice retreat was clearly visible. Finally, ice was present only on the shallow shelves. The Seaglider crossed regions with recent ice-melt and was even able to reach the entrance of the Norske Trough.

The data processing for these glider measurements was conducted at AWI. The first part consists of the Seaglider Toolbox from the University of Each Anglia; the second was exclusively composed for the data from the Greenland Sea.

The final hydrographic, position and drift data sets can be downloaded from
https://doi.org/10.1594/PANGAEA.893896.

**1 Introduction**

The Nordic Seas are shaped by a strong near-surface salinity contrast arising from the northward flow of saline Atlantic Water along their eastern rim in the Norwegian Atlantic Current (NwAC) and West Spitsbergen Current (WSC) and the southward flow of fresh Polar Water and sea ice along their western rim in the East Greenland Current (EGC) (Fig. 1). Due to strong cooling in winter, the Nordic Seas are one of the few regions in the World Ocean where convection normally reaches depths of 500 to 2000 m (Nansen, 1906; Rudels et al., 1989; Budéus

and Ronski, 2009). With this intermediate water ventilation it contributes substantially to the Atlantic Overturning Circulation (Schmitz and McCartney, 1993; Lumpkin and Speer, 2003). The convective overturning depends on the density stratification, which in the cold Nordic Seas is mostly set through salinity.

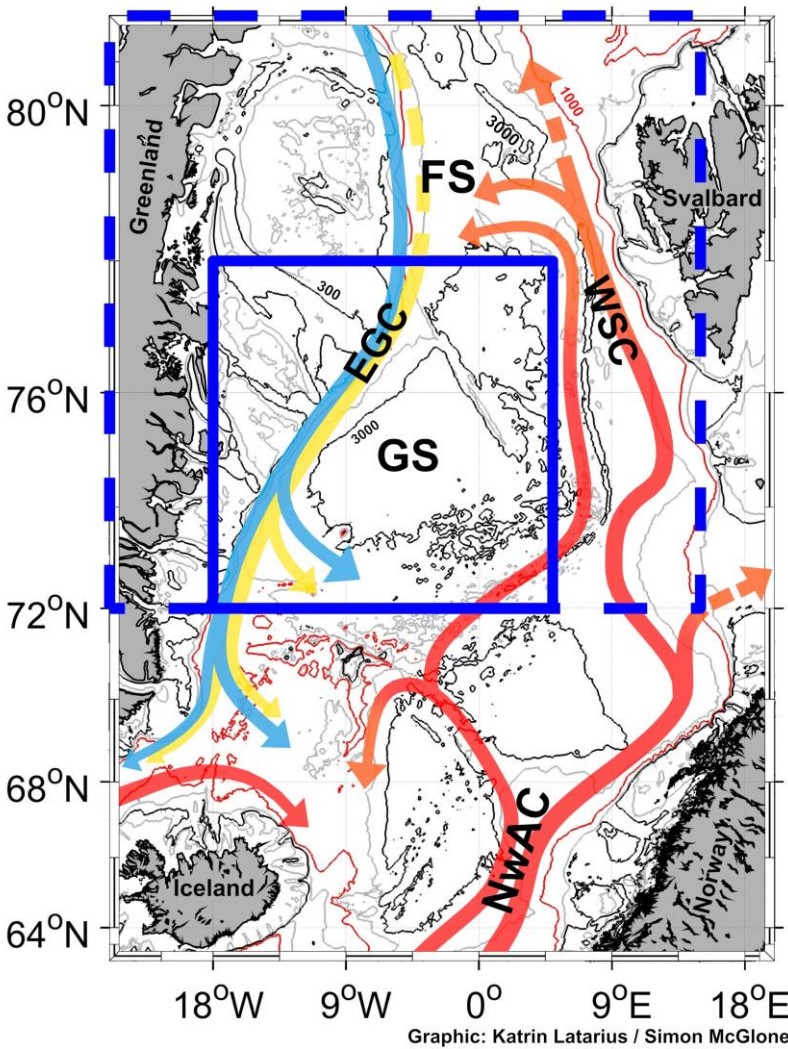

Graphic: Katrin Latarius / Simon McGlone


**Figure 1: The map shows the Nordic Seas. Topographic contours are given on the basis of RTOPO2 (Schaffer et al., 2016): the 1000 m contour is marked in red, the 3000 m and 300 m contours in black, and the 2000 m, 500 m and 200 m contours in gray. The inlet marked by the full blue line shows the area of Fig. 2, the inlet marked by the dashed blue line shows the area of Fig. 3.**

**Red to yellow arrows indicate the cooling of the warm and saline Atlantic Water as it flows through the Nordic Seas and Arctic Ocean. The blue arrows indicate the flow of cold and fresh Polar Water through the Nordic Seas.**
**EGC – East Greenland Current, WSC – West Spitsbergen Current, NwAC – Norwegian Atlantic Current, FS – Fram Strait, GS – Greenland Sea, NT – Norske Trough.**


Freshwater leaves the Arctic with the EGC in liquid form and as sea ice. For the liquid export de Steur et al. (2014) estimated 2100 km$^3$ yr$^{-1}$ over the period 2000-2010. The Fram Strait export of freshwater in sea-ice, averaged over the winters 2003-2008, is estimated to have been 2100 km$^3$ yr$^{-1}$ (Spreen et al., 2009). The annual average for 2000-2010 is 1900 km$^3$ yr$^{-1}$, when data gaps are filled using the average seasonal cycle (Hain et al., 2015). Haine et al. (2015) related these fluxes to other fluxes into and out of the Arctic as well as to the freshwater reservoir of the According to these considerations, liquid and sea ice fluxes with the EGC to the Nordic Seas account for almost 50 % of the total freshwater outflow from the Arctic for 2000 to 2010, with almost no changes in relation to the time span 1980 to 2000. Haine et al. (2015) expect an increase of the liquid outflow through Fram Strait by around 100 % for the next century, as at present the freshwater reservoir of the Arctic is increasing due to increasing river runoff and precipitation minus evaporation and due to ice melt. The sea ice outflow is expected to decrease due to the reduction of sea ice in the Arctic. This overall trend is anticipated to be superimposed by seasonal, interannual and decadal variability, mainly forced by variability in the wind-field (for a detailed discussion of wind-forced variability see Hain et al., 2015). Additional variability in the sea ice flux is introduced by the interplay of sea ice thickness, velocity and area (Smedsrud et al., 2011; Hansen et al., 2013; Spreen et al., 2009). Finally, large uncertainties are associated with this estimates, as the liquid freshwater flux, particularly the part close to the surface, as well as the different components of the sea ice flux are difficult to observe (Hansen et al., 2013; Spreen et al., 2009, de Steur et al., 2009; Hain et al. ,2015; and references included).

During late summer, low salinities were frequently observed in the near-surface layer of the deep basin of the Greenland Sea (GS) (Latarius and Quadfasel, 2016). However, this seasonal signal shows large inter-annual variability in magnitude and vertical extension. A fresh surface layer stabilizes the water column and may reduce wintertime convection in the GS (Latarius and Quadfasel, 2016). Oltmanns et al. (2018) observed the effect of slashed convection by a fresh surface layer for the Irminger Sea in winter 2010/2011. Very likely, the freshwater in the inner western Nordic Seas originates from the EGC, yet the explicit sources as well as the transport mechanisms are still unclear. De Steur et al. (2015) revealed local ice melt as the primary source for freshwater in the GS for summers 2011 and 2013. Dodd et al. (2009) found that a significant amount of sea ice leaves the EGC into the Nordic Seas whereas the liquid freshwater remains in the EGC up to Denmark Strait. However, it is also possible that liquid freshwater from the EGC reaches the inner western Nordic Sea. The transfer to the interior Nordic Seas may take place by eddies shedding off the Polar Front (Spall, 2011; Lherminier et al., 1999).

To investigate the spreading of freshwater from the western rim into the convection regions of the inner Nordic Seas, the project "Variation of freshwater in the western Nordic Seas" was conducted in the framework of the Research Group FOR1740: "A new approach toward improved estimates of Atlantic Ocean freshwater budgets and transports as part of the global hydrological cycle", funded by the German Research Association (DFG). The goal of this project is to capture and analyze fluctuations of freshwater in the western Nordic Seas and understand related processes. To achieve the necessary observations over several months with high spatial resolution two missions were conducted in the summer months of 2014 and 2015 in the western GS using autonomous gliders (Fig. 2). The gliders were operated in ice-free regions, but close to the ice edge. As glider navigation in shallow waters is difficult, the sections were limited to areas with water depth greater than 300 m.

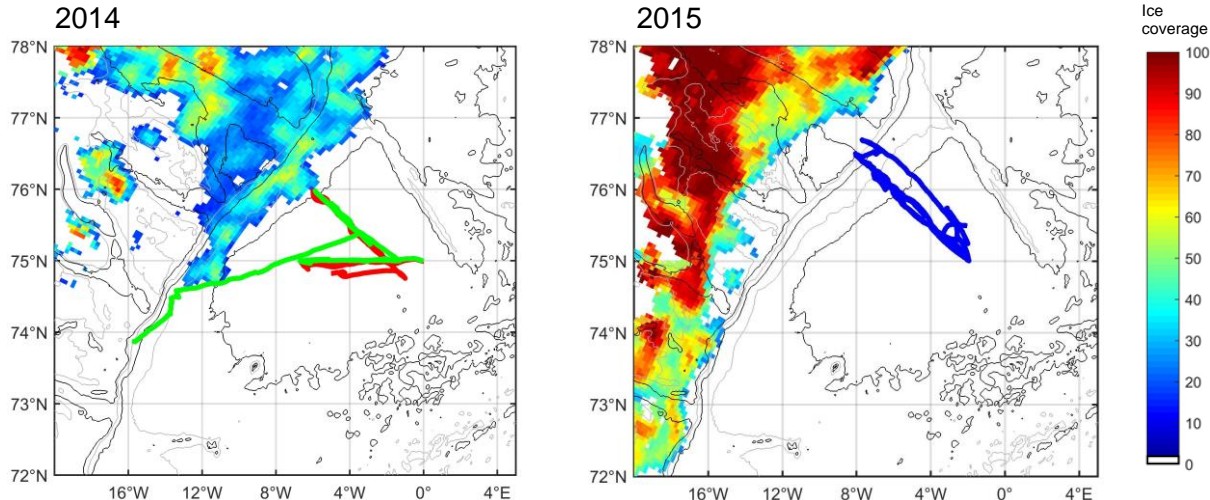

**Figure 2: Maps of the Seaglider missions in 2014 and 2015.**
**2014 (left): the thick red line shows the track of Seaglider 127 and the thick green line the track of Seaglider 558.**
**2015 (right): the thick blue line shows the track of Seaglider 127.**
**The thin black lines give water depth contours of 300 m, 1000 m and 3000 m as annotated for 2014 and the thin gray lines depth contours of 200 m, 500 m and 2000 m (not annotated) based on RTOPO2 (Schaffer et al., 2016). The color-coding denotes the concentration of sea ice in percent as derived from the sea ice data made available by DRIFT&NOISE (driftnoise.com); left: 2014/08/14, right: 2015/08/01. Details of the missions are summarized in Table 1. The development of the ice coverage during the missions time is described in detail in Section 2.4.**

In this paper we describe the details of the two missions in the challenging region near the ice edge and the shallow shelf east of Greenland (Section 2), the processing of the data and appendant uncertainties and error estimates (Section 3). In the last section, we give a brief description of the observations.

The hydrographic and drift data of both glider missions were published in the World Data Center PANGAEA.

**2 Seaglider Mission 2014 and 2015 in the western Greenland Sea**

**2.1 The western Greenland Sea**

The glider missions in summer 2014 and 2015 took place in the GS, which is the northernmost basin of the Nordic Seas (Fig. 1). The GS basin is up to 3600 m deep and flanked to the west by the steep continental slope east of Greenland. The EGC flows along the shelf break and the slope from the Arctic Ocean to the North Atlantic, transporting water masses of Arctic origin and sea ice. The West Spitsbergen Current, the northward extension of the Norwegian Atlantic Current, is flowing to the north along the eastern shelf break and slope, transporting mainly water of Atlantic origin. In Fram Strait part of the flow continues to the Arctic Ocean, while another part of the Atlantic Water recirculates and joins the EGC, thereby subducting below the Polar Surface Water (Quadfasel et al., 1987; Hattermann et al., 2016; von Appen et al., 2015).

Sea ice is transported with the EGC from the Arctic Ocean to the Nordic Seas. During winter, also local sea ice formation takes place in the western Nordic Seas. The ice primarily covers the western shelf and only in extreme winters reaches the deep GS (Comiso et al., 2001; Comiso et al., 2008; Comiso and Hall, 2014).

**2.2 Seagliders**

Seagliders are buoyancy-driven autonomous underwater vehicles that move through the water in a sawtooth pattern between the sea surface and a prescribed dive depth (Davis et al., 2003; Rudnick et al., 2004). Data are recorded during the dive and climb (down- and upward motion) and transmitted via satellite to the basestation during every surfacing. At that time, the glider can receive commands concerning its flight behavior and direction and its data sampling scheme. Typically, the glider is instructed by a target file, containing waypoints, about the planned courses of the mission, and by a science file about the sampling frequencies for the different sensors (see Table 1 and Section 3.1 with Table 2). New command files are sent if tuning of the flight behavior is needed.

For a given dive depth and dive time the glider's internal flight model calculates the needed buoyancy change and trim of the instrument for a sawtooth-pattern of down- and upward motion in direction to the next waypoint. The hydrodynamic shape and the small fins of the glider support the steering. The flight model additionally calculates the vertical velocity of the glider during dive and climb, which is used in the post-processing of the data. During every surfacing, the flight model compares the calculated position with the real one determined by GPS. From the discrepancy the depth-averaged current is calculated. If requested, these depth averaged currents can be used during the following dives in the flight model for an advanced calculation of the course to the next waypoint. If water depths less than the prescribed dive depth are expected, information from altimeter bottom tracking can be used for the ending of the downward motion.

The buoyancy of the glider is changed by changing the volume through inflation/deflation of an oil bladder (similar to profiling floats). The pitch (downward/upward orientation of the instrument) is changed by repositioning the center of mass by moving the battery pack forward/backward. Buoyancy and pitch together determine the angle of the downward or upward motion. To control the roll of the instrument an additional

weight is fixed axial asymmetric at the battery pack. As gliders behave like an "under-water sailplane", turning the battery to the right or left forces the glider to turn horizontally to the right or left accordingly.

Deep and slow dives need less energy and thus allow longer missions than shallow and fast dives; furthermore, they allow better steering between waypoints. On the other hand, shallower dives increase the horizontal resolution and faster dives allow capturing sections in shorter time. Seagliders were used in the described missions because high energy supply is characteristic for this type of gliders. The used instruments are restricted to work in ice-free water.

## 2.3 The missions

During summers 2014 and 2015, Seaglider missions were carried out in the western GS. The goal was to capture the spreading of freshwater from the western rim into the inner Nordic Seas. For this goal we run the glider(s) along a section between the deep GS basin and the EGC. Repeating the section in 2015 allowed observation of the variability of the spreading both during the course of the individual summers as well as between the two summers.

In 2014, the measurements started with an east to west section. Because of the ice coverage in the early summer, the mission had to be changed later to a southeast to northwest section (Fig. 2) perpendicular to the isobaths. For comparability, the southeast to northwest section was carried out in 2015 too (from 75° N/2° W to 76° N/6° W in 2014 and to 76°30' N/7°20' W in 2015). Only the last section conducted with glider 127 in 2015 was displaced to the north to capture also the Norske Trough. Table 1 summarizes information about both missions.

The focus of the project was on the near-surface hydrography and thus high horizontal resolution in the upper part of the water column was of foremost interest. Nevertheless, because of its unstable flight behavior during summer 2014 glider 127 was set to dive nearly always to 1000 m depth to achieve good steering. Glider 558, which had a more stable flight, was set for more than 70 % of the dives to 500 m depth. On August 21, glider 127 stopped diving, because the battery was almost empty (voltage-cutoff). The remaining energy was used to continuously send positions ashore. The instrument was recovered by MV Plancius on September 13. The energy of glider 558 lasted until the recovery with MV Ortelius at the shelf break of Greenland on September 4. During summer 2015, after new ballasting and trim by Kongsberg USA, glider 127 had a very stable flight. Thus, it was programmed to dive only to 500 m depth or to smaller depths on the shelf. There, altimeter bottom tracking kept the glider 50-100 m above the bottom.

## 2.4 The ice situation

Until middle of July, the ice situation in the observation site was quite similar in 2014 and 2015 (Fig 3): the broad shelf east of Greenland was covered with ice and the ice reached at least the position of the 1000 m depth contour. In the months after July, however, the ice coverage evolved differently in the two years.

In 2014, the ice between the perennial fast ice east of Greenland (Schneider and Budéus, 1995) and the central Fram Strait reduced continuously until September. Nevertheless, during the whole summer an ice tongue remained above the shelf break and the upper slope. Thus the ice edge, located above deep waters, was always

reached by the gliders at the northeastern-most position of the glider sections. However, the ice tongue prevented the glider to be operated across the EGC and to the shelf.

In 2015, a similar gap between the fast ice east of Greenland and the ice coverage in Fram Strait developed. However, in that year the shelf break and slope were completely ice-free from mid-July to mid-September. A large part of the shallow shelf was also ice-free, but was ice-covered again already in the beginning of September. This situation gave the opportunity to extend the sections to the shelf. With altimeter bottom tracking

a number of dives were carried out at around 300 m bottom depth. In August a more northern line across the shelf break to the inlet of the Norske Trough (Arndt et al., 2015) was executed. However, since the navigation of Seagliders in shallow water is problematic, the ice edge was never reached in 2015.

The glider missions in summer 2014 and summer 2015, give insight into the distributions of temperature and

205 salinity in the upper part of the water column. In summer 2015, the distribution was also observed in regions, where the ice coverage just disappeared. The observations can be interpreted in relation to the different ice coverage (see section 4).

210**Figure 3: The development of the ice cover in the western Greenland Sea during the time span of the glider missions in summer 2014 and summer 2015.**
**Left column for summer 2014, right column for summer 2015. Month and day of the individual ice concentration maps are given in the upper left corner. The maps are based on ice concentration data made available by DRIFT&NOISE (driftnoise.com). For each year, a sketch of the respective glider sections is added to the map (red**
215**lines and blue dots; the red dashed line in the bottom left map shows the track of glider 558 to the recovery position). Black contours give the 3000 m, 1000 m and 300 m depth contour based on RTOPO2 (Schaffer et al., 2016). The location of the map is shown as inlet in Fig. 1 with a blue dashed line.**

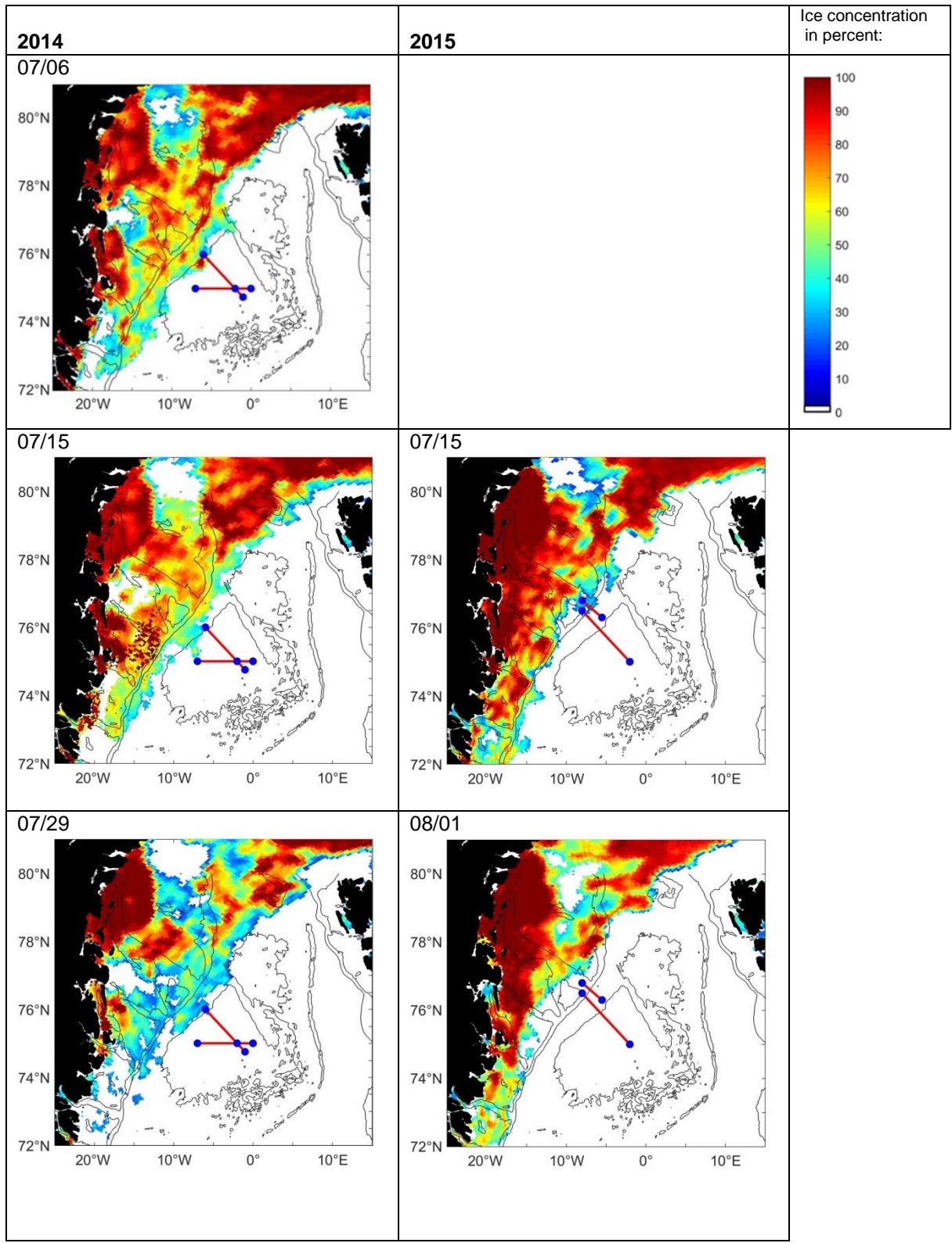

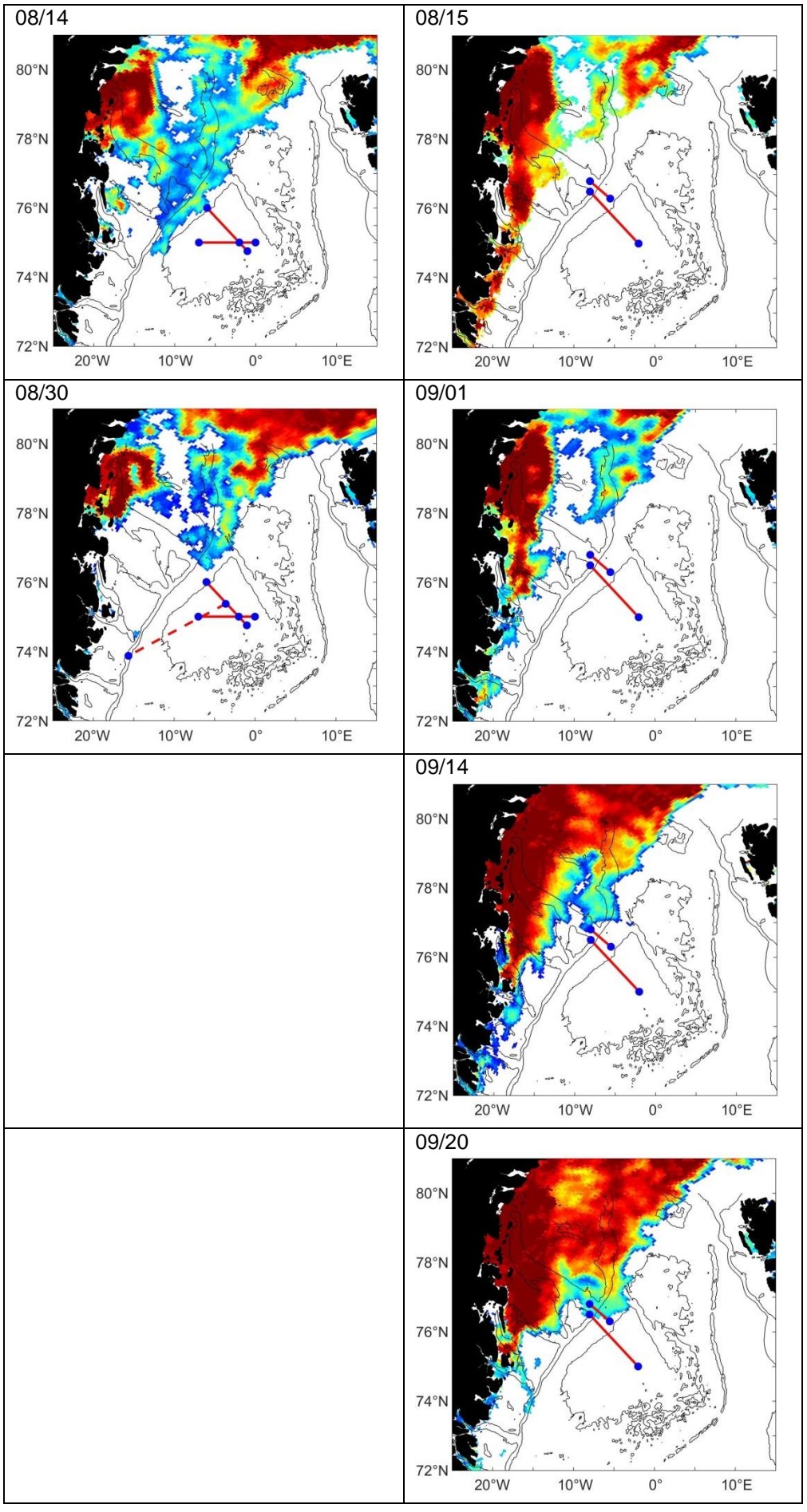

## 3 Data and processing

### 3.1 Glider setup, data transfer and raw data

The gliders were equipped with sensors for temperature, conductivity, pressure, oxygen (glider 127 only) and optical parameters (Table 2). However, during the missions, only temperature, conductivity and pressure data were recorded. Temperature and conductivity sensors have been calibrated by Sea-Bird (www.seabird.com) and the instruments were refurbished before the missions. The refurbishment included trimming and ballasting with tank tests and sea-trials. The communication with the gliders was performed via Iridium satellites and a basestation run by the company Kongsberg (www.km.kongsberg.com, https://usermanual.wiki/Document/ 4900039 BasestationUserGuide.3614056780.pdf). The piloting was carried out by a team in the section Polar Oceanography at the Alfred Wegener Institute, Helmholtz Centre for Polar and Marine Research (AWI) with support from Optimare, Harald Rohr.

During every surfacing, the data of the gliders were sent ashore. On the basestation, the data of each dive were decoded and transformed into two files containing the scientific data (eng file) and the technical data as well as information about the setting of the piloting parameters (log file), respectively. These files were the basis for the real-time analysis of the glider performance. If changes of the flight behavior were necessary, the pilots submitted new command files to the basestation, which were transferred to the glider during the next surfacing. Sometimes also new target files were sent for changes in the planned track, or new science files for changes in the sampling of the sensors. In addition, the data processing is based on the eng and log files.

### 3.2 Glider data processing

A glider measures temperature, conductivity and pressure while it is moving vertically and horizontally through the water. The relation between horizontal and vertical movement during our missions was 2:1 and an approximate localization of each measurement is possible with the start and end position of each dive. During the processing, the data were handled like ship-based CTD measurements consisting of temperature and conductivity reading related to ideally monotonously increasing pressure readings. Thus, the processing for these data basically follows the processing for ship-based CTD data. The aim of the processing is to eliminate random erroneous data, to correct systematic erroneous data and finally deliver profiles of temperature, conductivity, salinity and density on regular pressure steps. Expected systematic errors are a misalignment between temperature and conductivity measurements (time lag) (see for example Morison et al., 1994), a long-term distortion of the conductivity measurements (thermal lag) (Garau et al., 2011) and problems with the applicability of the sensor calibration, as the latter was conducted before the sensors were mounted on the instrument.

For a proper calculation of salinity and density, temperature and conductivity measurements from the same water body are needed. Otherwise spikes occur, especially if the glider was diving trough strong gradients. Due to different response times and different placement of the sensors on the instrument, different water bodies were measured by the temperature and conductivity sensors at the same time. This time lag was corrected by an alignment of temperature and conductivity data, which involves a vertical interpolation of the measured conductivity values.

The thermal lag, induced by the different geometry and the heat capacity of the cells, was corrected in a second step. This error lasts over several consecutive measurements. The correction was derived from the original data but it failed if extreme outliers were still present in the data.

Glider CTD data are more distorted than ship-based CTD data because

1. The vertical resolution of the glider measurements is low compared to that of a ship-based CTD. The gliders sample with 0.2 Hz (every 5 sec). With a typical vertical velocity of 0.1 to 0.12 m s$^{-1}$ this results in a vertical resolution of approximately 0.6 dbar (3-4 values in a layer of 2 dbar). A ship-based CTD samples with 24 Hz (every 0.04 sec) and is typically lowered at 0.5 to 1 m s$^{-1}$, resulting in a typical resolution of 0.02 dbar (48-96 values in a layer of 2 dbar).

2. The vertical velocity of a glider is much more variable than that of a lowered CTD, because the change in buoyancy (which accelerates the glider) is calculated with the flight model using a prescribed vertical density structure. The latter might deviate from the real local density structure. Due to the lower sampling rate, information about the vertical velocity of the instrument is also much coarser for gliders than for ship CTDs.  .

3. The water is not pumped through the glider's conductivity cell, because of limited energy resources; instead, the cell is freely flushed. Thus, the calculation of the flushing time depends on the uncertain vertical velocity (see 2.).

4. The quality of the interpolation for the time-lag correction depends on the vertical resolution, the information about the vertical velocity and the flushing time of the cell (see 1., 2. and 3., and Alvarez (2018) for numerical analysis of the performance of unpumped SBE sensors at low flushing rates).

5. The thermal-lag correction depends on the geometry and flushing time of the conductivity cell (see 2. and 3.).

6. The sensors are calibrated before mounting them on the glider, which limits the applicability of the calibration.

For these reasons glider data are much noisier and the time lag and thermal lag correction are of lower quality for glider CTD data than for ship-based CTD data. With respect to these problems, a data processing was set up at AWI, which consists of two parts (Table 3).

First, the raw data, as provided by the KONGSBERG basestation, were transformed to physical units and merged with the pre-mission calibration information. Corrections were applied to the data according to information from the gliders flight model and for the time lag and thermal lag of the sensors. This part was done by means of the UEA Seaglider Toolbox (UEA: University of East Anglia, Norwich, http://www.byqueste.com/toolbox.html). In addition to that, an extended processing was developed and applied to the glider data to exclude erroneous data, interpolate the data to discrete pressure levels, smooth the derived quantities and adjust absolute temperature and salinity to data from high-precision ship-based CTD casts close to the glider mission in space and time. An analysis was also made to determine if down- and upcast data showed systematic discrepancies and thereupon it was decided if both could be used or not. This second part of the processing involved knowledge of the regional hydrographic conditions. A prerequisite for a proper functioning of the

thermal lag correction is the exclusion of erroneous data/spikes. Thus, the first and second part of the processing

are entangled.

The individual steps of Table 3 are described in the following.

**A        UEA toolbox**

A.1    load and merge profile data and sg calib constants

The information from eng and log files (see Section 3.1) as well as from the sg_calib_constants-file was merged in a matlab-file. The sg_calib_constants-file contains the information about the calibration constants of the individual glider; sg – Seaglider.

A.2    calculate preliminary values

Preliminary values of the flight characteristics as well as temperature, conductivity and derived variables such as salinity and density and corrected pressure for each sensor were calculated. Based on the information about the calibration constants from the sg_calib_constants-file the frequency measurements for temperature and conductivity were transformed into physical units.

The movement, and thus the exact position, of the glider under water was derived from a flight model. These data were corrected by comparing the vertical velocity of the glider, calculated as change of pressure with time, with the flight model vertical velocity. The correction of the flight model has influence on derived variables such as dive-averaged currents, but also on the calculation of conductivity. In two steps various parameters were fitted so as to minimize the difference between the vertical velocities (Frajka-Williams et al., 2011).

A.3    vbd regression 1

(vbd: variable buoyancy device)

The maximum volume of the instrument was determined during pre-deployment tank tests and the reference environmental density was calculated from expected temperature and conductivity values for the mission region. Both values were adjusted to minimize differences between the flight-model vertical velocity and the observed vertical velocity.

A.4    vbd regression 2

The parameters of the hydrodynamic model were adjusted to minimize differences between the observed upward and downward velocities.

A.5    save changes in sg_calib_constants

During the vbd regression 1 and 2 (A.3 and A.4) some glider-specific parameters were changed. The changes were saved in the sg_calib_constants-file.

A.6    calculate preliminary values with new sg_calib_constants

See A.2; values of the flight characteristics as well as temperature, conductivity and derived variables such as salinity and density and corrected pressure for each sensor were recalculated after changes in the sg_calib_constants-file have been made.

Apply time-lag-correction and save thermal-lag correction

The time-lag-correction calculated from the UEA-toolbox was applied to the data. The thermal-lag correction was calculated from the UEA-toolbox but not applied to the data set at this step. Before a thermal lag correction erroneous data/spikes must be removed. The thermal-lag-correction was saved and applied at step B.3.

For details of the UEA toolbox see:

http://www.byqueste.com/toolbox.html

## B    AWI data processing

B.1    transfer of the hydrographic data from the UEA-toolbox output from matlab-structures to matrices

B.2    raw data inspection with gradient and min-max-criteria
       (+ individual corrections)

To eliminate spikes, data were deleted, when the difference between temperatures or conductivities of consecutive levels was larger than 0.25 °C or mS cm$^{-1}$, respectively. This gradient criterion was only applied below the thermo- or halocline.

Also all unrealistic data were deleted. The limits were temperatures lower than -2°C and higher than 15°C, conductivities lower than 23 mS cm$^{-1}$ and higher than 38 mS/cm. These limits were chosen on the background of local hydrography.

The mean vertical velocity (w) during the dives was between 10 and 12 cm s$^{-1}$. Lower velocities occurred at the start of the dive, in the apogee between down- and upward motion and at the end of the dive, but could have also occurred if the trim of the glider was wrong resulting in a slower vertical movement than normal. During these phases of low speed, the conductivity measurements can be wrong because of air bubbles in the water and insufficient flushing of the cell. Thus, data lines with vertical velocity smaller than 5 cm s$^{-1}$ were deleted.

B.3    thermal-lag correction with UEA toolbox output
       Calculation of salinity and density

The conductivity was corrected for thermal-lag according to the UEA toolbox output from A.6. Salinity and density were recalculated with the corrected conductivity.

Finally, pressure, temperature with time-lag correction, conductivity with thermal-lag correction and the derived quantities salinity and density were saved.

B.4    calculation of 2 dbar mean
       interpolation on 2 dbar levels
       (+ individual corrections)

To reduce the noise, the data were averaged within depth levels. Since we are interested in analyzing the distribution of freshwater in the near-surface layer with a typical thickness of 5 to 25 dbar, we chose 2 dbar  as the interval for calculation of mean values, which were then interpolated to discrete depth levels every 2 dbar from the surface to the dive depth. This vertical resolution was a compromise

between a sufficient vertical resolution and the reliability of the mean values (see Section 3.2, 1. for details)

In the final data set the variable NOBS gives the number of observations from which 2 dbar-means were calculated. If NOBS is empty for a certain line of data, values for temperature, conductivity, salinity and density were interpolated.

B.5 smoothing of density

iterative calculation of salinity with new density

calculation of conductivity with the new density and the new salinity

Salinity and density, calculated from the interpolated temperature, conductivity and pressure (B.4) were still very noisy. This was due to the lesser measurement accuracy of conductivity in relation to temperature, caused by the size of the conductivity cell. Particularly there were small instabilities in the density stratification, which we considered not to be real. Thus, density was filtered by a running mean over 11 layers (22 dbar). Afterwards salinity was iteratively changed in steps of 0.000065 until the respective calculated density reached the density from running mean within $\pm$ 0.000125 kg m$^{-3}$. The density threshold guarantees to reach the density value from running mean at least one order of magnitude more exact than the measurement accuracy. Finally, for data consistency, conductivity was recalculated from temperature, pressure and new salinity.

B.6 comparison of down- and up-casts

Since the CTD sensors are mounted on the top of the main body of the glider they are equally flushed during the down- and upward motion. Consequently, profiles from both directions can be used. This is different from ship-based CTD data, where only down-casts are used. Nevertheless, sometimes systematic discrepancies between down and up profiles have been reported (Garau et al., 2011). Fortunately, no systematic differences between down and up-casts were visible for any of the missions reported here. Thus, data from both casts were stored in the final data set. The direction of the cast is archived as the parameter direction: D: down, U: up.

B.7 adjustment of temperature and conductivity with ship-based CTD-data;

recalculation of salinity and density with new temperature and new conductivity

During the deployment of each glider, a ship-based CTD cast was carried out. The ship-based CTD temperature and conductivity data between 500 and 1000 dbar were compared with the mean temperature and conductivity profiles of all glider data in the same depth range within a spatial distance of $\pm$ 0.5 $^{\circ}$ in longitude and $\pm$ 0.25 $^{\circ}$ in latitude (i.e. approx. 30 km). Average differences in temperature and conductivity were calculated and all glider profiles were corrected by these offsets.

* individual corrections

The data processing steps listed above were not able to remove a number of errors, which were detected by visual inspection. In chapter 3.3.3 we describe how we dealt with these errors.

### 3.3 Quality of the data set – reasons for and effects of the different steps of the data processing

#### 3.3.1 CT sail specification:

Sea-Bird temperature sensor and free-flushed conductivity sensors, referred to as the CT sail, were installed on Seagliders 127 and 558. The CT Sail consists of three parts, the "CT" temperature sensor and conductivity cell, the temperature circuit board and the conductivity circuit board. These parts were disassembled and reassembled each time the CT Sail was calibrated and afterwards installed into the glider again. The calibration was conducted in advance to both missions. As the process of disassembly and assembly was not within the control of Sea-Bird, the applicability of the calibration is limited.

The specification of the CT sail:

  initial accuracy: conductivity $\pm$ 0.003 mS cm$^{-1}$, temperature $\pm$ 0.002°C

typical stability: conductivity $\pm$ 0.003 mS cm$^{-1}$ per month, temperature $\pm$ 0.0002°C per month

(comparable to SBE 37, Sea-Bird Electronics: www.seabird.com)

#### 3.3.2 Errors introduced by an insufficient time-lag correction:

(data processing step A.6)

Errors in the form of spikes in salinity occurred when the gliders moved through sharp gradients. The spikes were produced by insufficient alignment between temperature and conductivity in relation to pressure. Negative spikes were expected if the conductivity measurement was before the temperature measurements and positive spikes if conductivity lagged behind temperature (seabird software documentation: https://www.seabird.com/ cms-portals/seabird_com/cms/documents/training/Module12_AdvancedDataProcessing.pdf). In our glider profiles both kind of spikes were visible within one profile, reflecting variations in the vertical velocity of the glider. Thus, the systematic alignment of temperature and conductivity was not successful overall. Additionally the vertical resolution of the measurements did not enable an adjustment of the time lag correction. For ship-based CTD measurements typical adjustment is of approximately 20-30 msec for 50 cm s$^{-1}$ vertical velocity and 24 Hz sampling. This is equivalent to an alignment correction of 1 cm equivalent to 0.01 dbar for conductivity. To apply such a correction, the resolution of the original ship-based conductivity measurements has to be doubled by interpolation. If similar adjustment should be applied to the glider data, the vertical resolution would have to be refined with interpolation between measured values by a factor of 60 as on average the glider samples every 0.6 dbar. Especially in the region of sharp gradients we do not expect a good approach of the gradient by this refinement and therefore no improvement of the time-lag correction. Thus, we decided to leave these spikes uncorrected. Example spikes are shown for the glider 127 during the mission in 2015 in Fig. 4. They are of order $\pm$ 0.05 to 0.1 in salinity.

We decided to leave the decision how to deal with the spikes to the users of the data set. To help identification of affected profiles we list them in the Appendix. The spikes will possibly level out during gridding or averaging routines in further processing. For example, Queste et al. (2016) developed a method to deal with glider measurements across sharp gradients. They built composite profiles from the downcasts between the surface and the thermo-/halocline and from the upcasts between maximum depth and thermo-/halocline and combined these in a gridded data set.

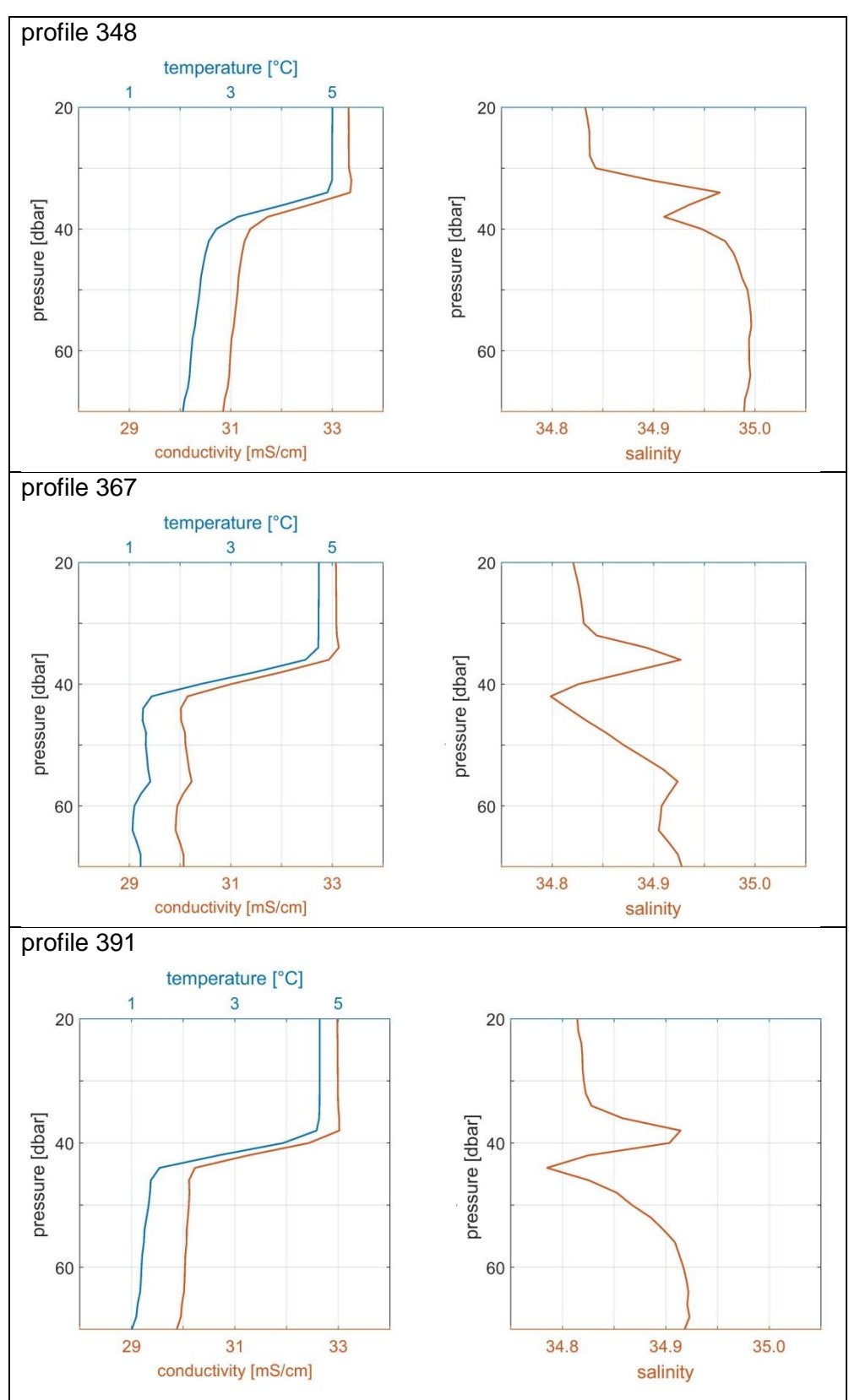

**Figure 4: Exemplarily, three temperature, conductivity and salinity profiles from glider 127 during summer 2015 demonstrate how spikes in salinity show up in the depth of highest vertical gradient between the surface layer and the water masses below. These spikes are generated by an insufficient time lag correction (see Section 3.3.2 for details).**


### 3.3.3 Visual inspection of the temperature, conductivity, salinity, density and vertical velocity profiles

(data processing step *individual corrections*)

By visual inspection of all individual profiles at different steps of the processing, several individual faulty values or profiles are detected:

- Spikes in salinity in the depth range of the thermo/halocline. These were removed, if they exceeded 0.1 (see Section 3.3.2)

- Wrong values during the apogee, which were not removed by the criterion w < 5 cm s$^{-1}$. These show up as temperature and conductivity values, which are far apart from the continuous profile, although the pressure did not change; they were removed.

- Outlier profiles of conductivity. Profiles, which are considerably separated from the entity of profiles of a mission, were removed.

- Profiles with large gaps in the depth of the largest gradient. If the gaps exceeded a depth range larger than the typical depth range of the thermo /halocline (> 10 dbar) the profiles were removed

- Incomplete profiles. When the dive was aborted by the glider-intrinsic software after an uncommanded change in the bleed counts of the vertical buoyancy device, these profiles were removed.

No individual temperature, conductivity or salinity values were removed, but always complete data lines or even the whole profiles were removed before the interpolation to 2 dbar levels took place. This results in a reduction of the original data sets between 2 % and 5% (Table 4).

### 3.3.4 Averaging the original measurements over 2 dbar intervals

(data processing step B.4)

As described above (Section 3.2) the original measurements were averaged over 2dbar to reduce the noise of temperature and conductivity. The 2 dbar mean values were on average based on 3 or 4 original data. Figure 5 shows the number of values for the averages and the standard deviations, exemplarily for glider 127 during the mission 2015 (for glider 127 and 558, mission 2014, the figures look similar).

There was a small difference in the number of records between the down- and upcast due to slightly different velocities. The reduced numbers at the maximum depth of the profiles and at the beginning of the downcast reflect the rejection of data during the start of the dive and during the apogee/bottom dead point, where the vertical velocity was below 5 cm s$^{-1}$. Larger numbers at the surface reflect measurements before the profile was started.


The standard deviations of temperature and conductivity below 200 dbar depth are small (0.01 °C/ 0.01 mS cm$^{-1}$) but much higher especially between 30 and 40 dbar (0.12 °C/ 0.12 mS cm$^{-1}$). The high standard deviations reflect strong temperature and salinity gradients at shallow depths. Near the surface, the standard deviation of conductivity is particularly high (up to 0.38 mS cm-1). This can be due to air bubbles entering the conductivity

cell (Section 3.2, 3.). Unrealistic values have been excluded *(Section 3.3.3).

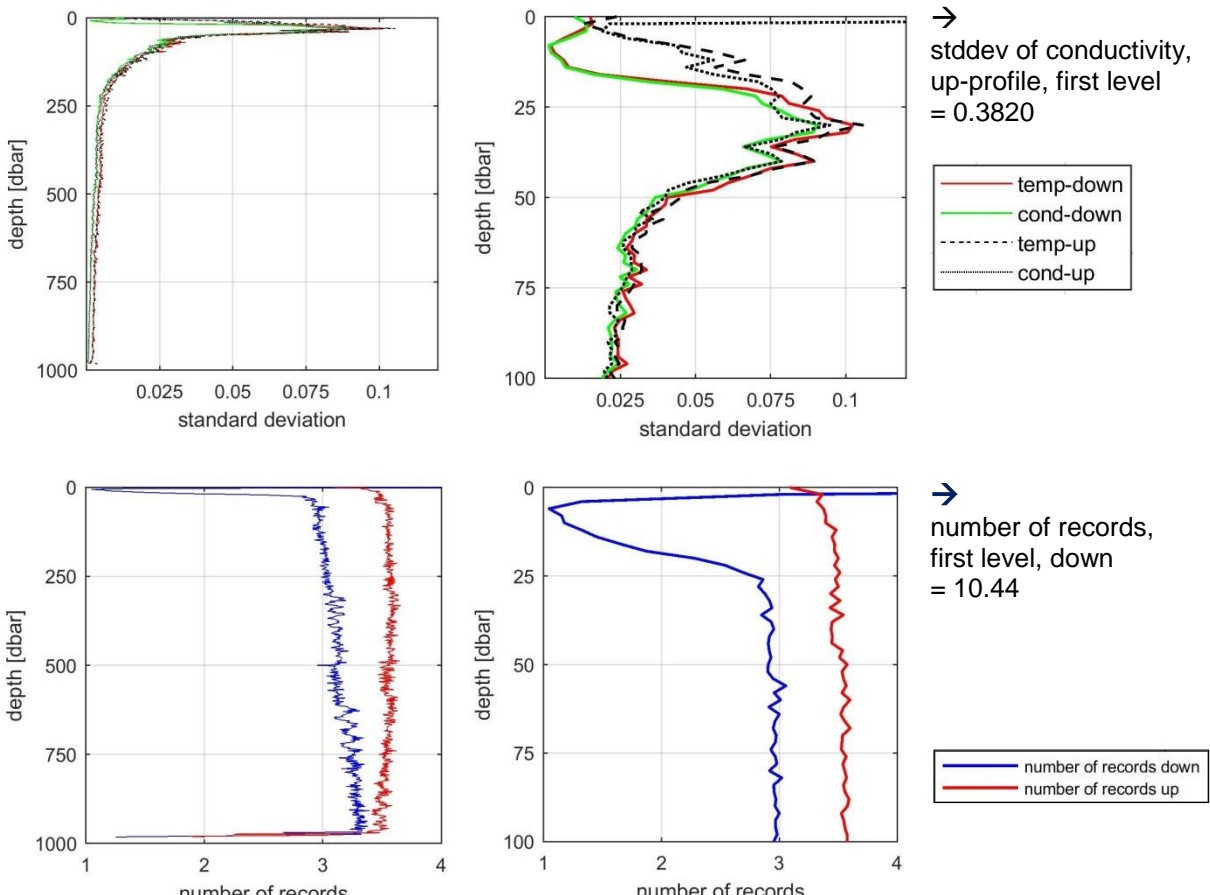

**Figure 5: Standard deviation (top) and number of original data (bottom) for 2 dbar averages of temperature and conductivity are shown for the down- (red) and upcasts (green). Glider 127 during the mission 2015 was used as an example. On the left the total profile is shown, in the middle only the upper 100 dbar of the profiles. Extreme values at the surface and the legend for the figures to the left are given in the right column.**

### 3.3.5 Smoothing of density and salinity

(data processing step B.5)

To quantify the noise reduction resulting from step B.5, where the criterion of stable density was applied, we calculated the variability of a profile before and after the step. The variability is defined here as the difference between consecutive values of salinity in a profile. Figure 6 shows the variability for all individual salinity profiles; again, glider 127 during the mission 2015 was used as an example. In the upper 80 dbar the noise reduction is of order 10 %, but between 45% and 63% below. The average reduction for the whole depth range is 22 % for glider 127 in 2014, 13 % for glider 558 in 2014 and 33 % for glider 127 in 2015.

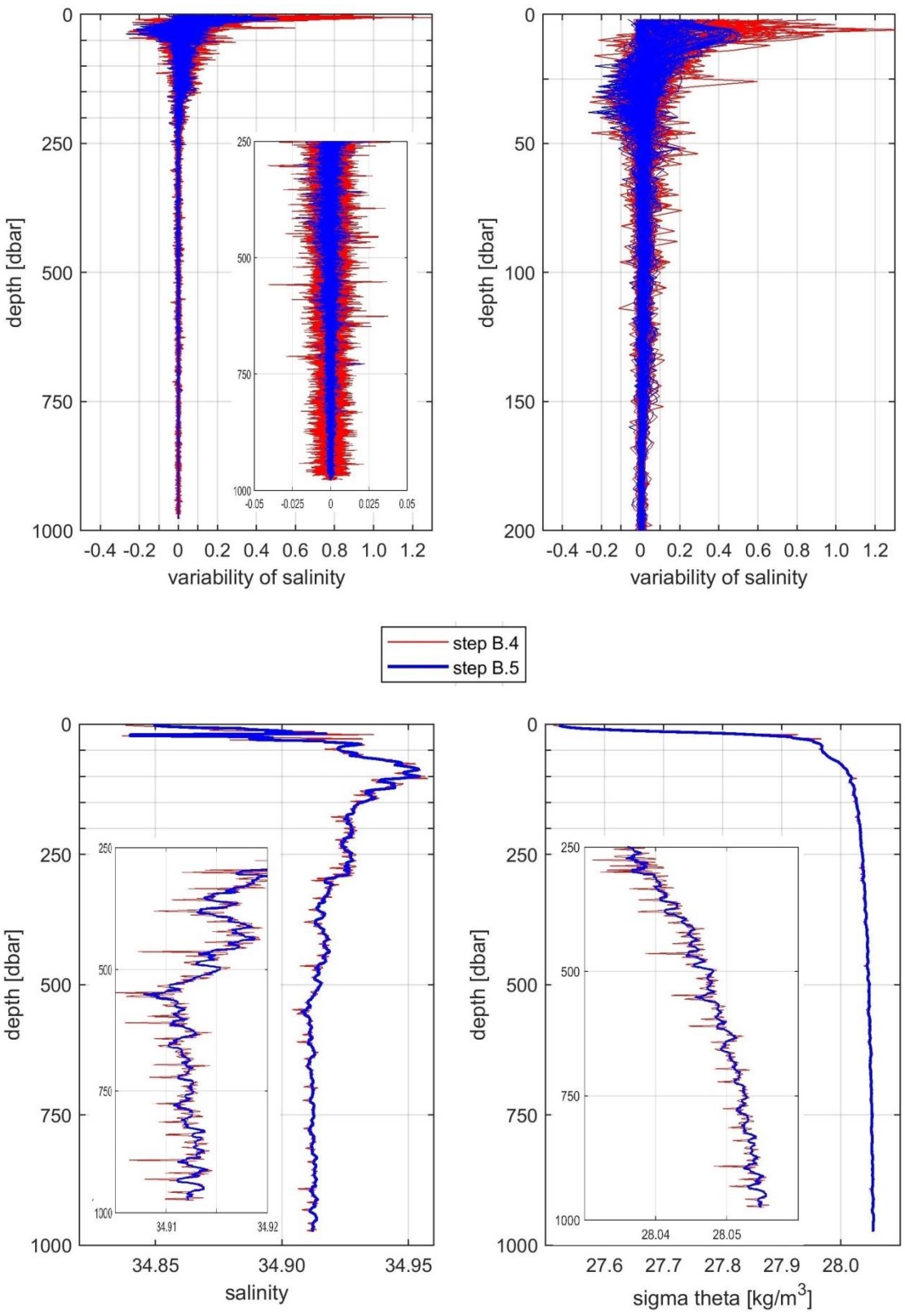

**Figure 6: (Top) Variability for all individual salinity profiles before (red) and after (blue) smoothing of density, (left) for the total profile depth, (right) for the upper 200 dbar; again, glider 127, mission 2015 was used as an example. (Bottom) an individual salinity (left) and density (right) profile before (red) and after (blue) smoothing of density. Extracts of 250 to 1000 dbar are inserted.**

### 3.3.6 Adjustment of absolute values to ship-based CTD data

(data processing step B.7)

The adjustment of absolute values to ship-based CTD data was at least an order of magnitude larger than the accuracy of an SBE 37 and thus demonstrates the urgent need of high-quality ship-based CTD data in the vicinity of a glider mission.

Values for the different inaccuracies are summarized in Table 4.

Possibilities and constraints for using the glider data presented here are briefly discussed at the end of section 4.

### 4 Distribution of temperature and salinity from glider missions

This section provides a brief description of the temperature and salinity distributions measured during the two summer glider missions.

Typical hydrographic conditions in the GS reflect the major circulation features in the Nordic Seas. The deep Greenland Basin is bounded by the EGC on the western side. Cold and fresh Polar Surface Waters are transported with the current from the Arctic Ocean into the subpolar North Atlantic. On the eastern side, the Greenland Basin is bounded by the West Spitsbergen Current. Waters of Atlantic characteristic – warm and salty – flow along the Norwegian Coast and shelf break to the North and undergo cooling on the way. In Fram Strait, part of the Atlantic water continues into the Arctic and part recirculates (Rudels et al., 2005; Hattermann et al., 2016). The Recirculated Atlantic Water joins the EGC and partwise subducts below the Polar Surface Water (see Fig. 1) (Rudels et al., 2002). The central GS is dominated by Arctic Intermediate Waters, which are relatively cold and salty (Blindheim and Rey, 2004; Rudels et al., 2005; Rudels et al., 2012). The deep basin of the GS is filled with this water mass, formed by local convection during winter. A very weak stratification is characteristic for the inner GS, reflecting the winter convection. Strong seasonal variations are only observed on top of the Arctic Intermediate Water. The near-surface layer is dominated by summer heating/winter cooling. Additionally during late spring and summer occasional freshening is observed (de Steur et al., 2015; Latarius and Quadfasel, 2016).

Our observations with the gliders captured the Arctic Intermediate Water in the central GS and the Polar Surface Water as well as the Recirculating Atlantic Water near the ice edge in the west, thus confirming the typical hydrographic conditions. However, only in 2015 the core of the EGC with cold (below -1°C) and fresh (salinity between 34 and 32) waters down to approximately 100 m was reached (Fig. 7). In 2014, the ice conditions did not allow to extend the sections that far to the west (see Section 2.4). The West Spitsbergen Current in the east was never reached.

The glider observations give insight into the interannual variability close to the surface. The warm near-surface layer was around 20 m thick in 2014 but up to twice that thick in 2015 (see Fig. 7). We expected that due to a much more extended ice coverage the summer heating started later in 2014 (see Section 2.4). However, the most obvious difference between the summers 2014 and 2015 was the near-surface salinity distribution. In summer 2014 waters with very low salinities (31-33) reached up to 3°W, hence occupied two thirds of the section. These waters were restricted to the upper 10-15 m (see also Fig. 8-10) and only in the north-west, close to the ice edge

low salinities were accompanied by low temperatures. This kind of water most likely was the remnant of locally melted ice (de Steur et al., 2015).

Also in 2015 the lowest salinities in the near-surface layer were observed at the northwestern end of the section. However, the signature was different. The water was restricted to the western end of the section and was not as

fresh as in 2014 and the freshwater was not that concentrated near the surface. As described above it reflects the Polar Surface Water flowing with the EGC from the Arctic Ocean through Fram Strait to the south and continuing its way along the shelf break to the subpolar North Atlantic (Rudels et al., 2005). Also the development of the ice coverage during summer 2015 (Fig. 3), suggests that this water mass was not a signature of recent ice melt. Since beginning of August, the shelf break was not covered with ice. The ice retreated to the

shallow shelf close to the coast until mid-September.

The presented distributions of temperature and salinity, measured along sections from the inner GS to the EGC during summer 2014 and summer 2015, show signs of freshwater intrusions close to the surface. The development within a single summer as well as the interannual differences are demonstrated. The freshwater

intrusions are not masked by the inaccuracies of the measurements, as we described in detail in Section 3, as the absolute difference between the Polar Surface Water and the Arctic Intermediate Waters is of order 4-6 K for temperature and 2-4 for salinity. For further analyses, one has to take into account that in opposite to ship-based CTD sections, glider sections are never "quasi-synoptic". Thus, the combination of low time resolution and high spatial resolution provided by glider measurements must be considered, when deriving quantitative conclusions

from the observed distributions.




For Figures 7 to 10 all profile data of the final data set along a specific section were gridded in the horizontal at 0.05° longitudinal resolution (approximately 1.3 km). In the vertical, the profiles were already interpolated to 2 dbar levels during the data processing.

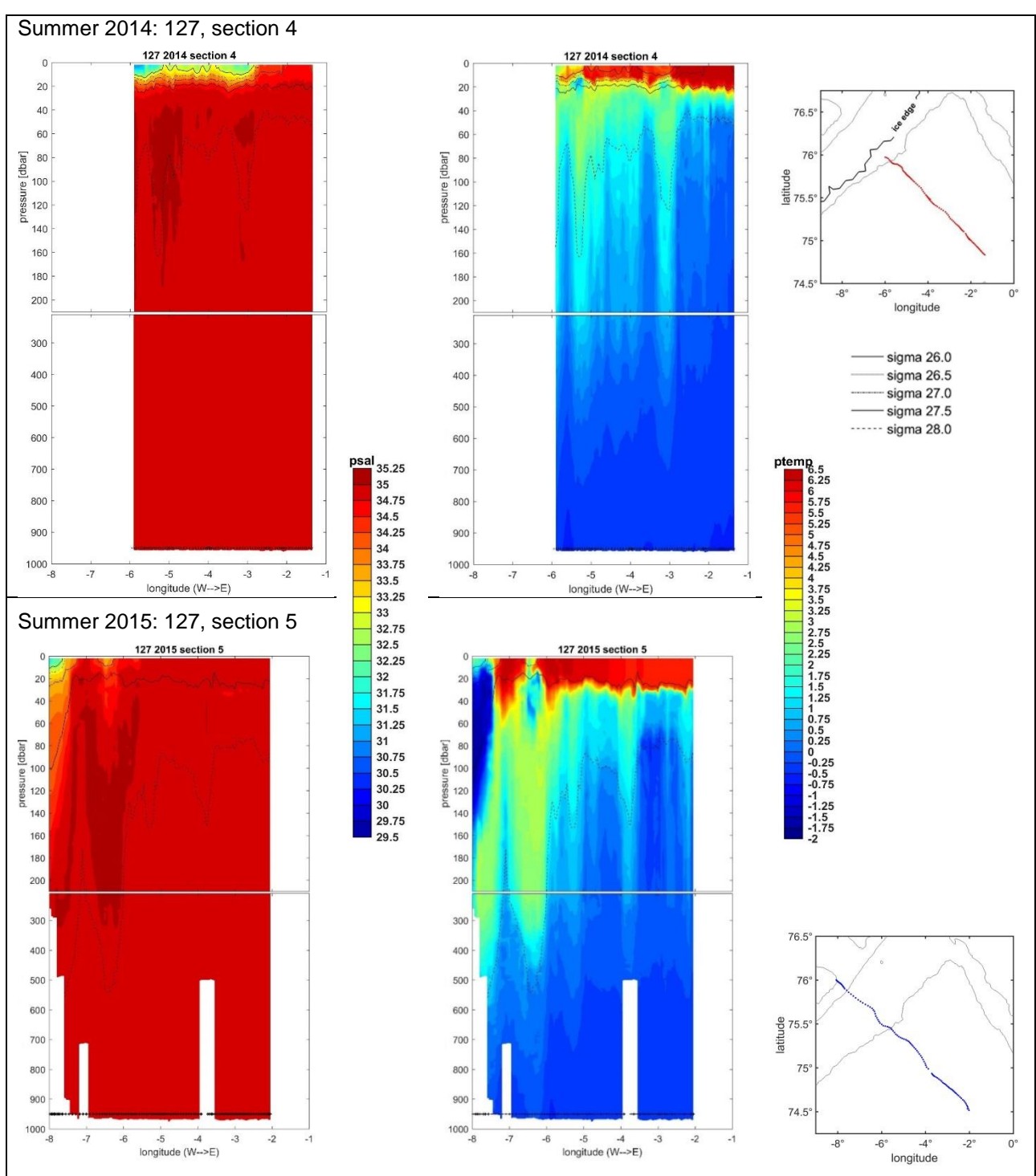


**Figure 7: Deep sections (0 to 1000 dbar), one for each summer as an example. Left column: salinity, middle column: potential temperature, top: 2014, bottom: 2015. In the right column, map extracts show the position of the sections. For 2014, also the ice edge at the arrival time of the glider at the edge is included in the map. During 2015, no ice was observed within the map extract.**

**Sigma-contours for 26.0/26.5/27.0 and 27.5 kg m⁻³ are superimposed.**

**The stars at the bottom of each section mark the position of the profiles before the gridding of the data in the horizontal took place.**

**Section 4 from glider 127, 2014, from Northwest (left) to Southeast (right); original dives: 168 to 280, profiles 260 to 353 of the final data set, time span August 2 to 8, ice edge information from August 5.**

**Section 5 from glider 127, 2015, from Northwest (left) to Southeast (right); original dives: 144 to 197, profiles 231 to 336 of the final data set, time span August 11 to 22.**

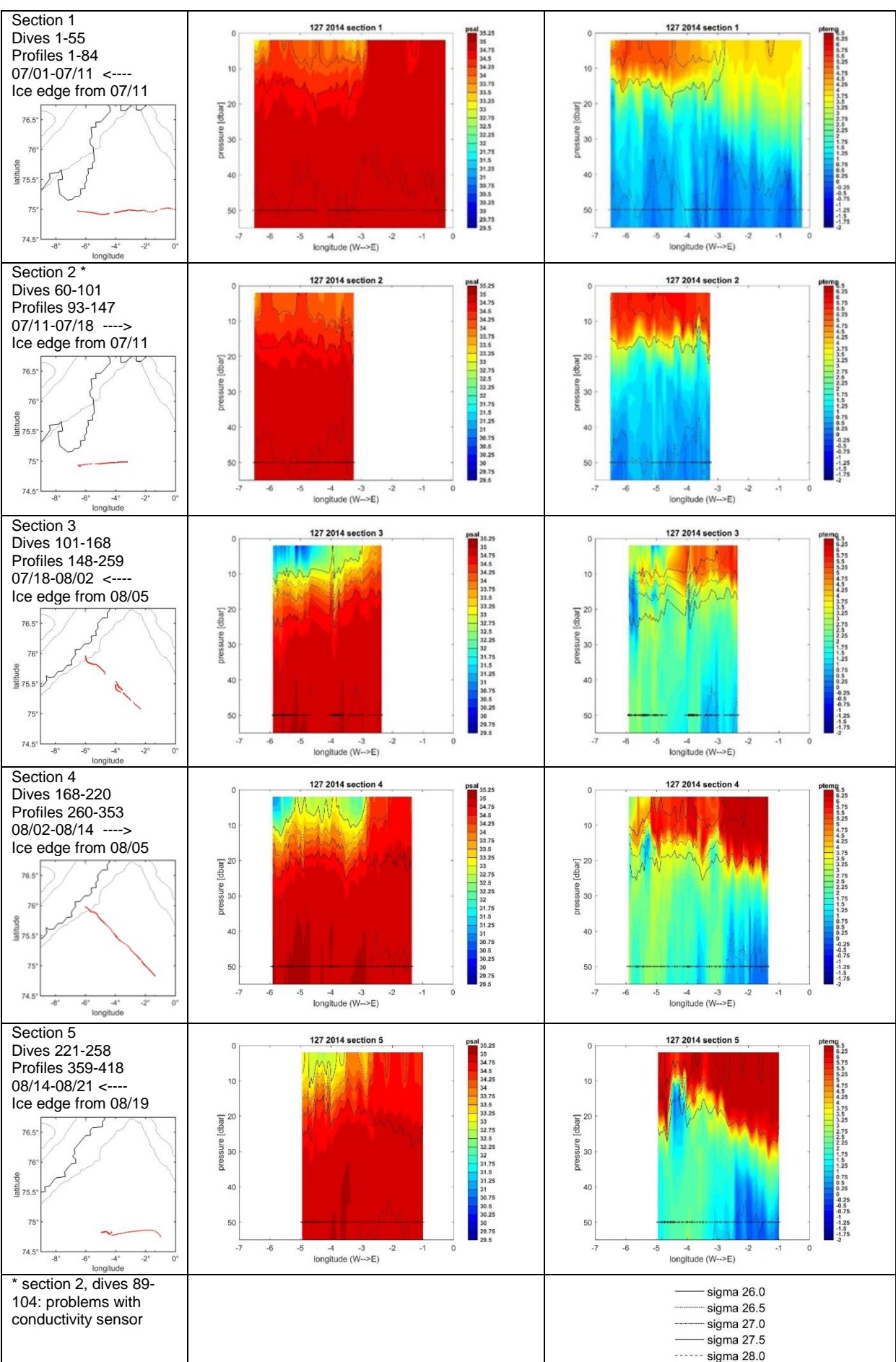

Section 1
Dives 1-55
Profiles 1-84
07/01-07/11  <----
Ice edge from 07/11

Section 2 *
Dives 60-101
Profiles 93-147
07/11-07/18  ---->
Ice edge from 07/11

Section 3
Dives 101-168
Profiles 148-259
07/18-08/02  <----
Ice edge from 08/05

Section 4
Dives 168-220
Profiles 260-353
08/02-08/14  ---->
Ice edge from 08/05

Section 5
Dives 221-258
Profiles 359-418
08/14-08/21 <----
Ice edge from 08/19

* section 2, dives 89-104: problems with conductivity sensor

sigma 26.0
sigma 26.5
sigma 27.0
sigma 27.5
sigma 28.0

**Figure 8: Sections: 2014, glider 127, sections 1-5. Shown are the upper 55 m of the water column.**
**Left column: information and map extract with location of the profiles and ice edge for the time span of the section (see Section 2.4 for details); numbers of original dives on the section, numbers of profiles from the final dataset on the section, time span, and direction of the section: <--- east to west, ---> west to east.**
**Middle/right column: salinity/ potential temperature. Both plots are overlaid with the sigma-contours 26.0, 26.5, 27.0, 27.5 and 28.0 kg m$^{-3}$ (see legend for line style in Fig. 7). Stars at 50 m depth mark the position of the profiles on the section before the gridding of the data in the horizontal took place. For section 2 many profiles are missing because of problems with the conductivity sensor.**

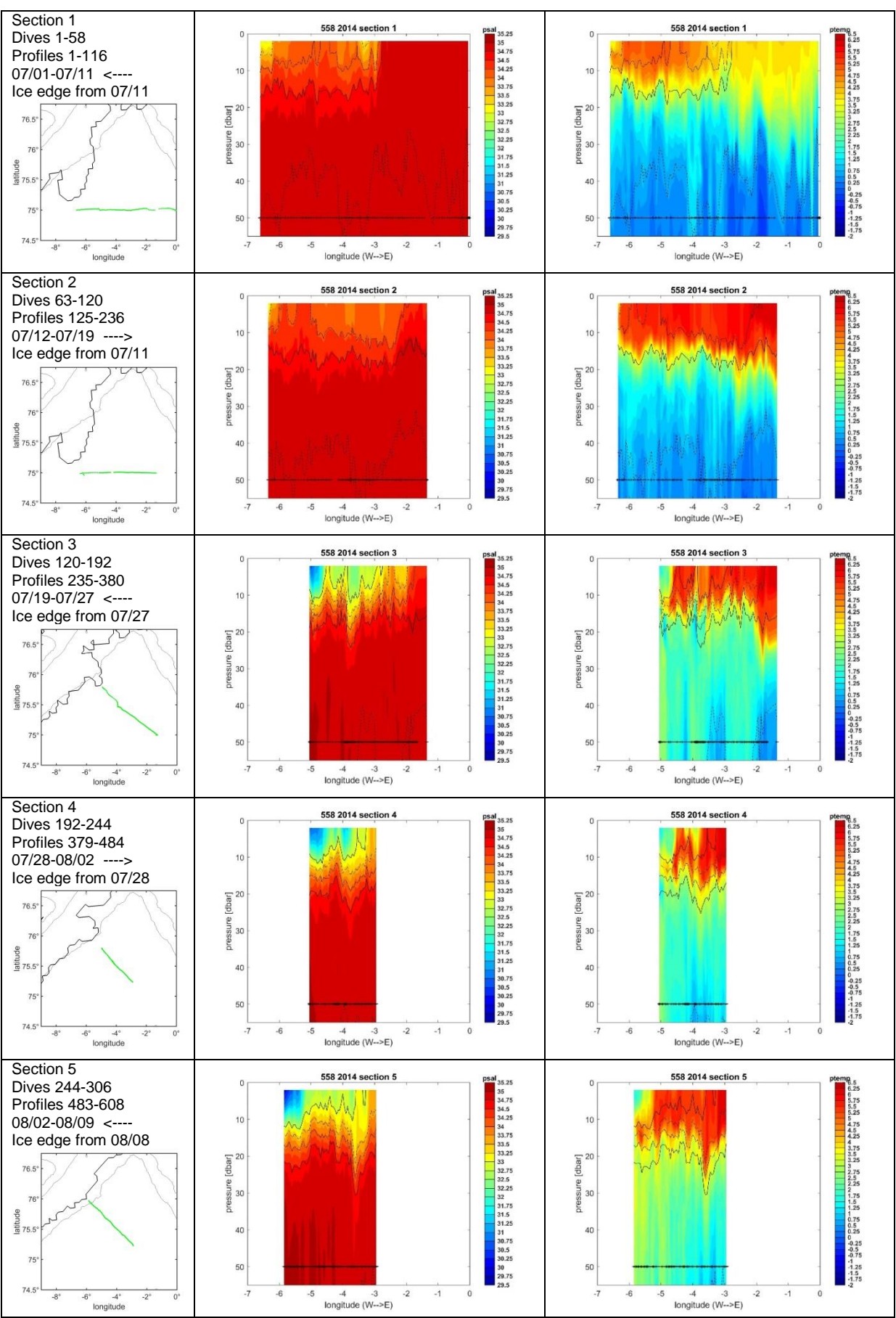

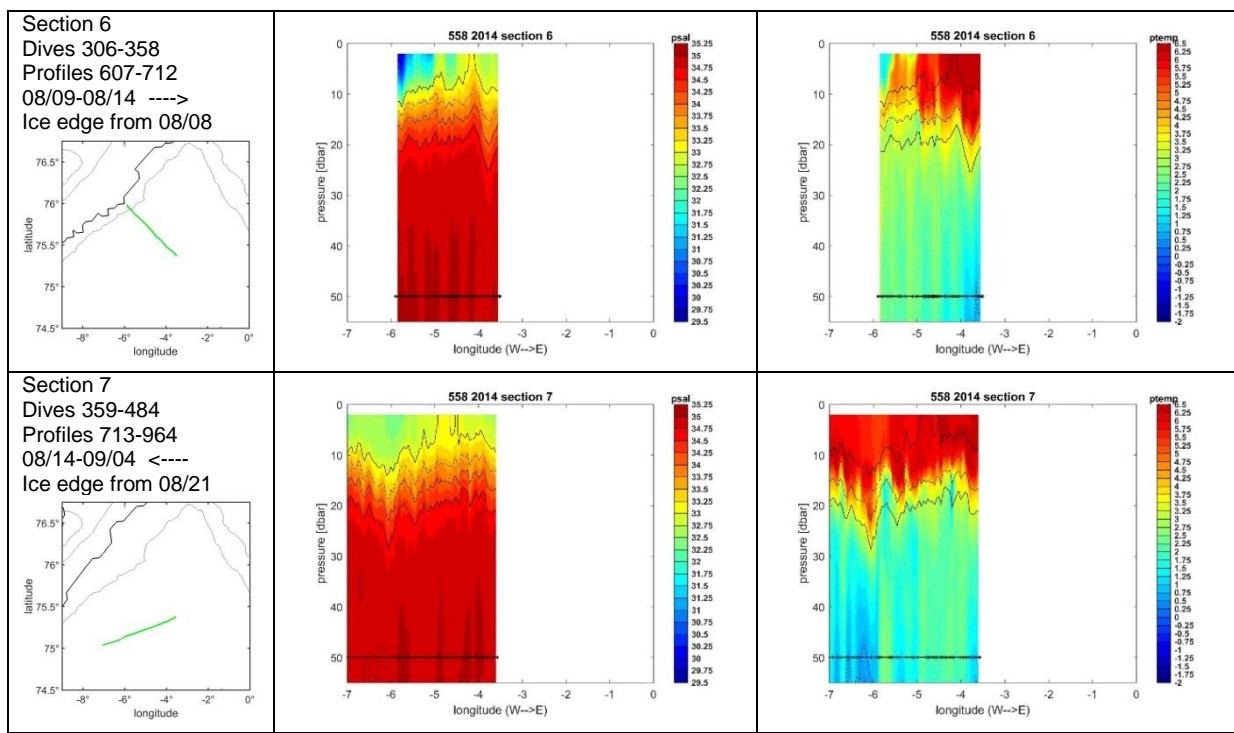


**Figure 9: Sections: 2014, glider 558, sections 1-7. Shown are the upper 55m of the water column.**
**Left column: information and map extract with location of the profiles and ice edge for the time span of the section (see Section 2.4 for details); number of original dives on the section, number of profiles from the final dataset on the section, time span, and direction of the section: <--- east to west, ---> west to east.**
**Middle/right column: salinity/ potential temperature. Both plots are overlaid with the sigma-contours 26.0, 26.5, 27.0, 27.5 and 28.0 kg m$^{-3}$ (see legend for line style in Fig. 7). Stars at 50 m depth mark the position of the profiles on the section before the gridding of the data in the horizontal took place.**

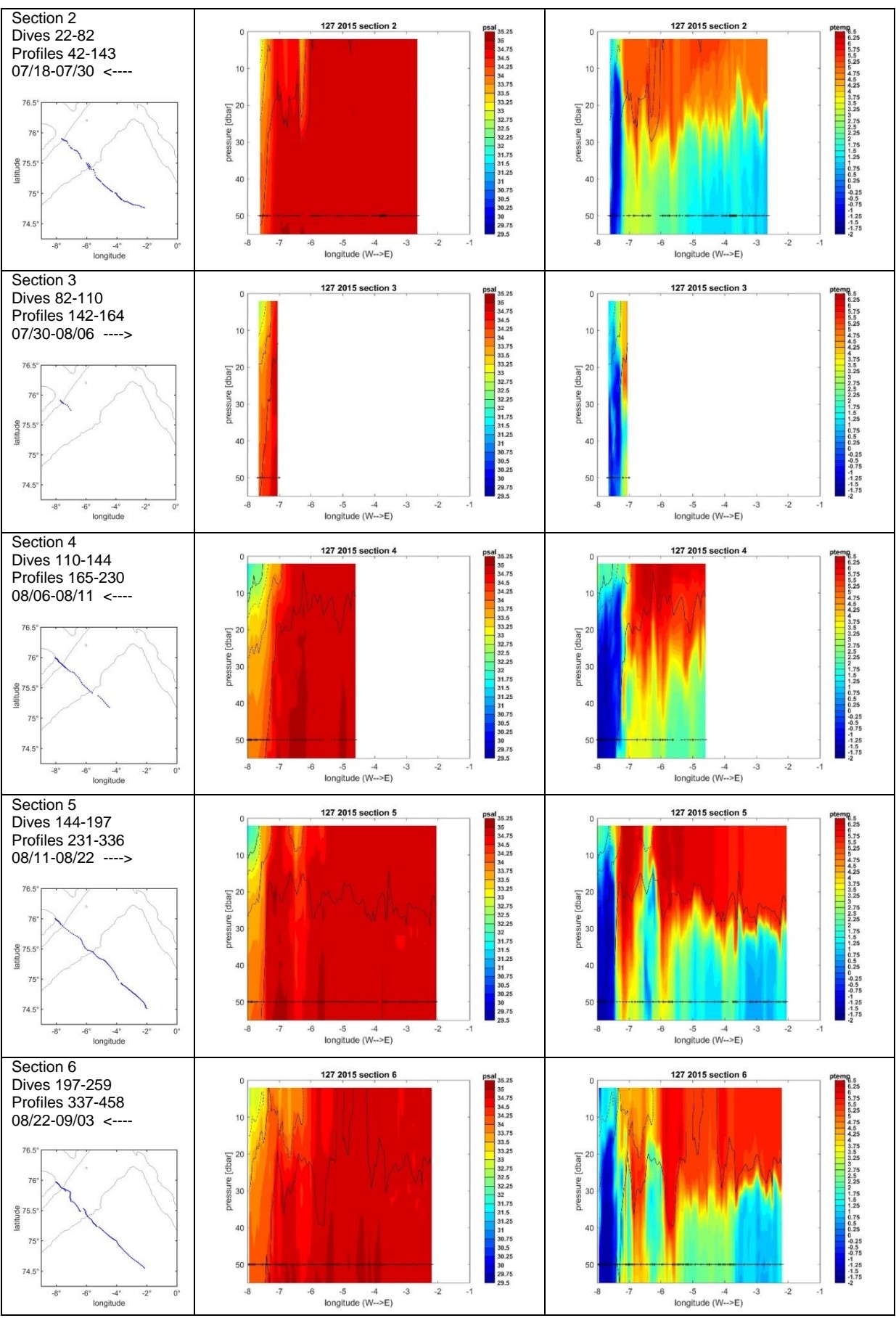

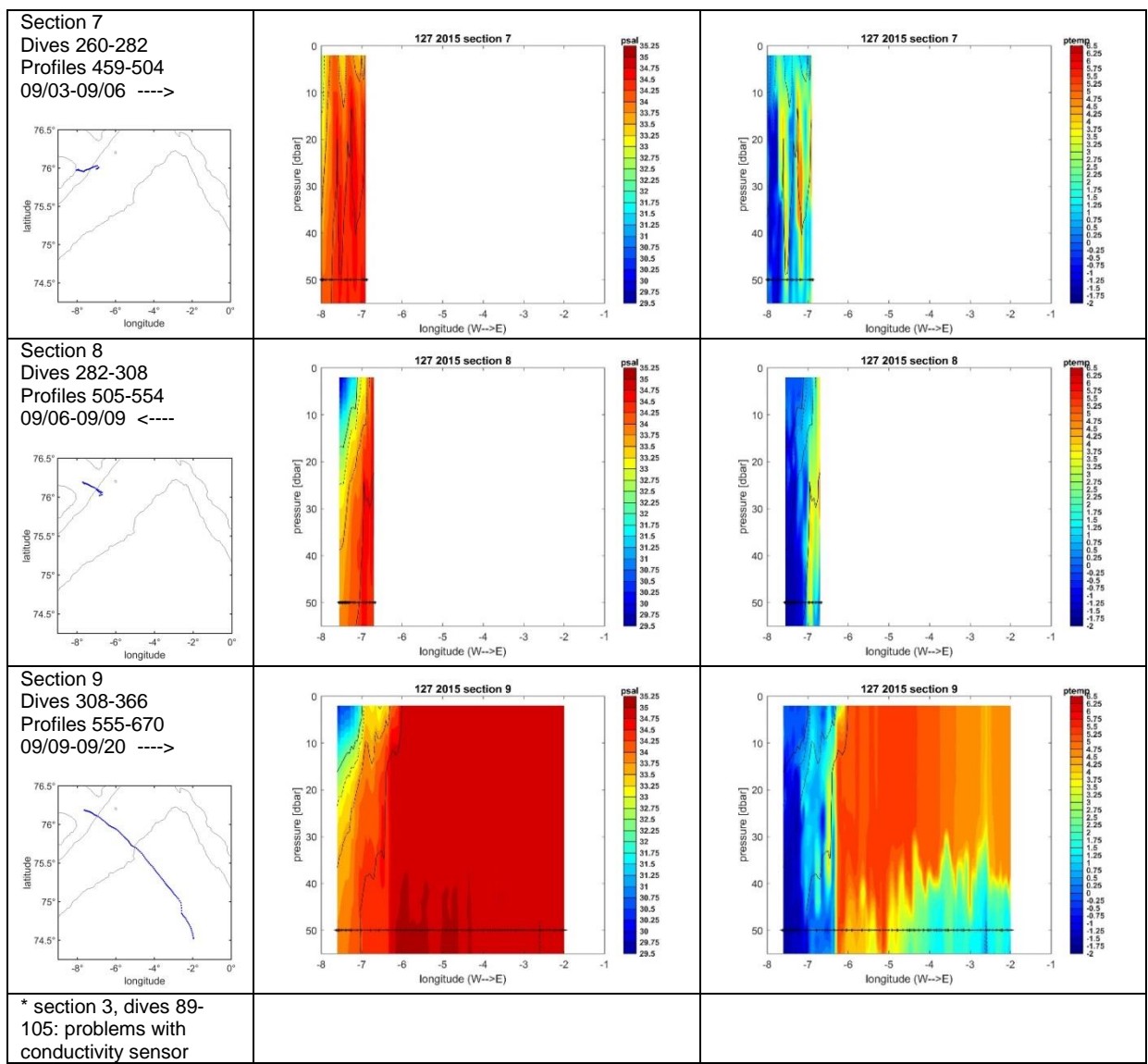

**Figure 10: Sections: 2015, glider 127, sections 1-9. Shown are the upper 55 m of the water column.**

Left column: information and map extract with location of the profiles (no ice was observed during the whole mission
time in the map extract); number of original dives on the section, number of profiles from the final dataset on the
section, time span, and direction of the section: <--- east to west, ---> west to east.

Middle/right column: salinity/ potential temperature. Both plots are overlaid with the sigma-contours 26.0, 26.5, 27.0,
27.5 and 28.0 kg m^-3 (see legend for line style in Fig. 7). Stars at 50 m depth mark the position of the profiles on the
section before the gridding of the data in the horizontal took place. For section 3 many profiles are missing because of
problems with the conductivity sensor.

**5 Data availability**

The glider data are available from the PNAGAEA data archive. The digital open access data library PANGAEA is a publisher for earth system science and hosted by the Alfred Wegener Institute. The glider data are stored in

two ASCII *.txt files and two netcdf *.nc files for each glider and mission, one file for the drift data and one for the hydrography data.

Main link: https://doi.org/10.1594/PANGAEA.893896.

Sub-links: https://doi.org/10.1594/PANGAEA.893725 for glider 127, summer 2014, hydrography;

https://doi.org/10.1594/PANGAEA.893734 for glider 127, summer 2014, drift;

https://doi.org/10.1594/PANGAEA.893730 for glider 558, summer 2014, hydrography;

https://doi.org/10.1594/PANGAEA.8937237 for glider 558, summer 2014, drift;

https://doi.org/10.1594/PANGAEA.8937256 for glider 127, summer 2015, hydrography;

https://doi.org/10.1594/PANGAEA.8937235 for glider 127, summer 2015, drift.

**Appendix**

List of individual profiles with spikes in the thermo/halocline.

For details see Section 3.3.3.

Glider 127 2014:

Dive no: 10-13, 17, 11, 24, 76, 82, 206-208, 212-214, 220-227, 229-231, 233-234

Glider 558 2014:

Dive-no: 1, 3-13, 15-25, 85-86, 91-93, 101-103, 110-112, 116-121, 125-127, 390

Glider 127 2015:

Dive-no: 2-7, 9-17, 19-32, 34-67, 75-77, 106-107, 109-115, 117-124, 167-226, 230, 233, 329-420.

The dive-no is named *observation number* in PANGAEA ASCII *.txt files.

**Competing interests**

The authors declare that they have no conflict of interest.

**Acknowledgments**

Katrin Latarius was funded by the German Research Association (DFG) within the framework of the Research Group FOR1740: "A new approach toward improved estimates of Atlantic Ocean freshwater budgets and transports as part of the global hydrological cycle".

We thank Bastian Queste from the University of East Anglia for developing and making the UEA toolbox publicly available and supporting us in using it.

During the mission piloting we got prompt, detailed and friendly support from Harald Rohr, OPTIMARE, KONGSBERG Germany as well as from KONGSBERG USA.

During the missions, Thomas Krumpen and his colleagues from DRIFT&NOISE provided high-resolution ice
information.

We would like to thank captain and crew of RV POLARSTERN and MV ORTELIUS and the Company
OCEANWIDE EXPEDITION for glider deployment and recovery under rough sea conditions.

We thank two anonymous reviewers for their comments, which helped considerably to improve our manuscript.

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

**Table 1: General information on the two glider missions in summer 2014 and summer 2015**

| year | 2014 | 2014 | 2015 |
|---|---|---|---|
| glider SN | 127 | 558 | 127 |
| deployment | 2014/06/30<br>75°00' N, 0°00' E/W<br>with RV Polarstern (PS85) | 2014/07/01<br>75°00' N, 0°01' W<br>with RV Polarstern (PS85) | 2015/07/14<br>75°45' N, 3°08' W<br>with RV Polarstern (PS93.1) |
| recovery | 2014/09/13<br>74°31' N, 1°58' W<br>with MV Plancius * | 2014/09/04<br>73°45' N, 16°16' W<br>with MV Ortelius * | 2015/10/05<br>75°45' N, 3°08' W<br>with RV Polarstern (PS94) |
| E-W sections | dive 1-101 | dive 1-120 | |
| SE-NW sections | dive 101-220 | dive 120-358 | dive 1-420<br>northern sect: dive 282-343<br>shelf-dive 128-151, 257-270,<br>shelf-dive with altimeter bottom<br>tracking: 292-316 |
| | | to recovery position (NE to SW)<br>dive 359-484 | |
| | 2014/08/21 voltage-cutoff;<br>surface drift until recovery;<br>position and drift data for "dive"<br>221-258 are available | | |
| total | 220 dives<br>52 days<br>179 dives to 1000 m<br>41 dives ≤ 500 m<br>typical distance for<br>500/1000 m dives: 2.8/3.8 km<br>total mission distance: 910 km | 484 dives<br>65 days<br>142 dives to 1000 m<br>342 dives ≤ 500 m<br>typical distance for<br>500/1000 m dives: 2.1/4.0 km<br>total mission distance: 1266 km | 420 dives<br>84 days<br>500 m ≤ 329 dives ≤ 1000 m<br>91 dives ≤ 500 m<br>typical distance for<br>500/1000 m dives: 1.7/4.8 km<br>total mission distance: 1678 km |

* We gratefully acknowledge the support by OCEANWIDE EXPEDITIONS.

**Table 2: Origin of the gliders 127 ad 558 and their setup during the two missions in the Greenland Sea in summer 2014 and summer 2015.**

| glider sn | 127 | 558 |
|---|---|---|
| manufacturer<br>year of delivery | University of Washington<br>2006 | iROBOT<br>2012 |
| missions presented here | 2014, 2015 | 2014 |
| sensors | conductivity, temperature: SBE CT sail, SN 0050, unpumped<br>pressure: Druck PCDR 4020, SN 2438976<br>chlorophyll, CDOM, scattering: Wet Labs Optics, SN BB2fVMG-163<br>oxygen: Aanderaa Optode AA 3830, SN: 11<br>altimetry: Applied Acoustic Engineering Seabed Transponder 955, SN 021/761 | conductivity, temperature: SBE CT sail, SN 0190, unpumped<br>pressure: Paine 211-75-710-05, SN 269511<br>chlorophyll, CDOM, scattering: Wet Labs Optics, SN BB2FLVMT-87<br>altimetry: Applied Acoustic Engineering Seabed Transponder 955, SN 283/2444 |
| sampling | 0 – 1000 m: conductivity, temperature and pressure every 5 sec<br>(Wet Labs Optic data and oxygen data were not sampled) | 0-1000 m: conductivity, temperature and pressure every 5 sec<br>(Wet Labs Optic data were not sampled) |

**Table 3: Flow chart of glider data processing with UEA toolbox and AWI additions. For each individual profile, the eng and log files contain the scientific and technical data. The sg_calib_constants-file contains the information about the pre-deployment calibration of the individual gliders.**
**sg_calib_constants – file containing calibration information, vbd – vertical buoyancy device, UEA – university of East Anglia, T- temperature, S – salinity, C – conductivity.**

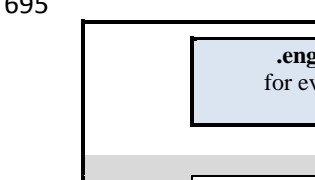

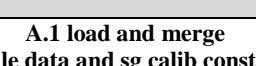

| | |
|---|---|
| **.eng/.log files** for every profile | **sg_calib_constants** for the individual mission and glider |

**UEA TOOLBOX**

**A.1 load and merge** profile data and sg calib constants

**A.2 calculate preliminary values**

**A.3 vbd regression 1**

**A.4 vbd regression 2**

**A.5 save changes in sg_calib_constants**

**A.6 calculate preliminary values with new sg_calib_constants** apply time-lag correction and save thermal-lag correction

**UEA-toolbox output:** eng/log structure, hydrography-structure, flight-structure, gps position, date, calibration, ..

**AWI DATA PROCESSING**

**B.1 transfer of hydrographic data to matrices**

**B.2 raw data inspection with gradient and min-max-criteria ( + individual corrections)**

**B.3 thermal-lag correction with UEA toolbox output calculation of S and density**

**B.4 calculation of 2 dbar mean interpolation on 2 dbar levels ( + individual corrections)**

**B.5 smoothing of density interative calculation of S with new density calculation of C from new density and new S**

**B.6 comparison of down- and upcast**

**B.7 correction of T and C with CTD-data recalculation of density and S with new T and new C**

**final data set for the individual mission and glider**

**Table 4: Summary of the data quality and effects of the different steps of the data processing for the three individual data sets (glider 127 mission 2014, glider 558 mission 2014 and glider 127 mission 2015). References to the specific steps of the data processing are given in the left column.**

700

| | glider 127 mission 2014 | glider 558 mission 2014 | glider 127 mission 2015 |
|---|---|---|---|
| specification of CT sail (see Section 3.3.1) | initial accuracy: conductivity $\pm$ 0.003 mS cm$^{-1}$ temperature $\pm$ 0.002 °C typical drift: conductivity $\pm$ 0.003 mS cm$^{-1}$ per month temperature $\pm$ 0.0002 °C per month | | |
| spikes not eliminated by time-lag-correction (see Section 3.3.2) | $\pm$ 0.05 to 0.1 in salinity | | |
| data reduction with gradient/min-max criterion and individual corrections (step B.2 and B.4, see Section 3.3.3) | 5 % | 2.2 % | 4 % |
| standard deviation of 2 dbar averages (step B.4, see Section 3.3.4) | temperature 0-80 dbar 0.06 °C 80-1000 dbar 0.004 °C 0-1000 dbar 0.009 °C  conductivity 0-80 dbar 0.05 mS cm$^{-1}$ 80-1000 dbar 0.002 mS cm$^{-1}$ 0-1000 dbar 0.007 mS cm$^{-1}$ | temperature 0-80 dbar 0.06 °C 80-1000 dbar 0.003 °C 0-1000 dbar 0.007 °C  conductivity 0-80 dbar 0.05 mS cm$^{-1}$ 80-1000 dbar 0.003 mS cm$^{-1}$ 0-1000 dbar 0.007 mS cm$^{-1}$ | temperature 0-80 dbar 0.05 °C 80-1000 dbar 0.006 °C 0-1000 dbar 0.009 °C  conductivity 0-80 dbar 0.06 mS cm$^{-1}$ 80-1000 dbar 0.004 mS cm$^{-1}$ 0-1000 dbar 0.008 mS cm$^{-1}$ |
| number of records for 2 dbar averages: | 3.5 | 3.2 | 3.3 |
| Average variability* of salinity in the surface layer, below, and in the whole depth range before step B.5 $\rightarrow$ after step B.5, and variability reduction in percentage.  (step B.5, see Section 3.3.5) *variability is defined as the difference between consecutive values of salinity in a profile (see Section 3.3.5) | 0-80 dbar 0.051 $\rightarrow$ 0.046 $\rightarrow$ 10 % 80-1000 dbar 0.0019 $\rightarrow$ 0.0007 $\rightarrow$ 63 % 0-1000 dbar 0.0058 $\rightarrow$ 0.0045 $\rightarrow$ 22 % | 0-80 bar 0.073 $\rightarrow$ 0.067 $\rightarrow$ 8 % 80-1000 dbar 0.0011 $\rightarrow$ 0.0006 $\rightarrow$ 45 % 0-1000 dbar 0.0069 $\rightarrow$ 0.0060 $\rightarrow$ 13% | 0-80 dbar 0.028 $\rightarrow$ 0.025 $\rightarrow$ 11 % 80-1000 dbar 0.0025 $\rightarrow$ 0.0013 $\rightarrow$ 48 % 0-1000 dbar 0.0046 $\rightarrow$ 0.0031 $\rightarrow$ 33 % |
| comparison with ship-based CTD (step B.7, see Section 3.3.6) diff_T/C/S= temperature/conductivity/salinity difference between glider and ship-based CTD | diff_T=-0.0266 °C diff_C= -0.0104 mS cm$^{-1}$ (diff_S=0.0163, not used) | diff_T=-0.0095 °C diff_C= -0.0063 mS cm$^{-1}$ (diff_S=0.0024, not used) | diff_T=-0.0389 °C diff_C= -0.0316 mS cm$^{-1}$ (diff_S=0.0025, not used) |