# Peer review of "Near-ice Hydrographic Data from Seaglider Missions in the Western Greenland Sea in Summer 2014 and 2015"

_Earth System Science Data, 2018_

## Referee Comment (RC1) · Anonymous Referee #1 · 16 Jan 2019

GENERAL COMMENTS

This paper presents data, and associated data processing, from two Seagliders operating during the summer of 2014 and 2015 in the Western Greenland Sea. The multi-year seasonal context of the dataset is of value to the scientific community and this should be emphasized with more specific detail throughout the extent of the paper. The processing steps are numerous but are well documented. The final dataset appears to only contain post-processed data (original values are not preserved). The addition of quality flags would assist future users in determining the level of processing that has been applied, and help to identify known erroneous values (ie salinity spikes). Data

accuracies and statistics should be discussed more quantitatively within the text portion of the paper, with particular attention given to any potential limitations and future applications of the data. There are typographical and grammatical errors throughout the paper that require attention. I suggest the following revisions be addressed prior to resubmission.

SPECIFIC COMMENTS

Abstract (P1): The abstract states the goals of the field program but lacks mention of the scientific findings and/or why the associated dataset is of relevance to the scientific community.

L32 (P1): For consistency in the sentence that refers to figure 1, add "in West Spitsbergen Current (WSC)" after "along their eastern rim".

Figure 1 caption: I suggest the acronym definitions get moved to the bottom of the figure caption, below the description of the red arrows. Additionally, L48 (P1): consider rewording this sentence. The gradient in the red arrows is what is indicative of the cooling, so this should be stated first. Suggested rewording: "The red to yellow arrows indicate the relative cooling of the warm, saline Atlantic Water as it flows through the Nordic Seas and Arctic Ocean."

L52-56 (P2-3): This paragraph should be more descriptive and/or reorganized. A few brief clarifying statements could help. The first sentence lacks temporal context (are the numbers listed a general average? Are the numbers listed percentages of total output over a year, during a specific season, or?). You then mention the temporal variability in freshwater transport in the EGC; how does this variability relate to the EGC's contribution to total freshwater output (the 50-75% you mention)? Additionally, variability in freshwater transport can be due to either variability in current speeds and/or variability in total volume fraction of freshwater to total seawater. Which (or both) is more influential here?

L140-142 (P5-6): The parenthetical description of "roll" does not help to clarify this sentence. "Turn to the left/right" could refer to roll or yaw. Either eliminate, or use the actual axis of rotation in your description.

Section 2.3 (P6): Table 1 is very thorough and contains a lot of good information. You should summarize more of the pertinent information in the text portion of this section. Consider moving the last paragraph of section 2.2 to section 2.3. Your first explanation of the reasoning behind the mission planning is too general (L146-147); include more specifics up front. L154: "but later concentrated on a southeast to northwest section" – why? Also, please explain the voltage-cutoff and unstable flight behavior during summer 2014 (glider 127). What were the causes? It appears these issues were resolved for the second deployment of glider 127 but this is not well explained.

Section 2.4 (P7): Your description of the differences in ice regimes between the two seasons is well detailed. What are the implications to the datasets in terms of their capabilities, comparability with each other and with other datasets, etc?

Figure 3 (P8-9): On P9 why is one of the glider tracks represented by a dotted line but this is not described in the figure caption?

L228 (P10): Explain how the glider data are comparable to CTD data (in sample frequency, resolution?).

L280-281 (P12): Describe your matchup criteria (spatial and temporal) quantitatively (what does "close" mean?). I see the spatial criteria is listed in B.7 but should be mentioned here (or B.7 referenced).

B.4 (P15): Is the variable 'numrec' the same as 'NOBS[#]'? The latter is what is used in the dataset available on PANGEA. This final column in the published dataset has missing values for select entries (ie PS93_SG127_hydrography profile 348, 8dbar, Direction 1). What does this mean? How can there be < 1 obs used for a line entry, when all other fields are populated?

B.5 (P16): The steps outlined here seem circular. Why not smooth conductivity? Do you have a reference for this method and can you describe why you chose the thresholds that you did? What is the mean difference between original and recomputed values?

L333-335 (P18): Why did you choose not to include quality flags in your dataset, especially since you chose to leave uncorrected spikes? It seems this should be indicated in the dataset for the user in some way, either as a header note or (preferably) quality flags.

L395 (P21) and Figure 6: Your final statement of a reduction in salinity variability of 50% is too vague. It really only applies to Figure 6, glider 127, mission 2015 in the deep layer (which is difficult to see in the plot). Your plot should better exemplify this (zoom in on deep layer instead of surface?), and text should be more descriptive (is it an average reduction, and what is the std of the difference in variability?). Similarly, in table 4 'variability reduction of salinity' should better describe the where the numbers came from, I assume they are averages of the differences in variability at each depth interval? I see an average reduction of mean variability of between ∼8-10%, ∼30-60%, and ∼13-48% in the three layers, respectively, and across all missions.

L407-409 (P22): Great.

Table 4 (P24): Given the data quality and accuracies you outline in table 4, what are the limitations (are there any?) to use of the dataset, in terms of better understanding fluctuations/dynamics of changing freshwater fluxes in Nordic Seas?

Section 4: Interesting observations stemming from this dataset; the multi-year span provides great context for comparisons. This section lacks any concluding remarks.

TECHNICAL CORRECTIONS

L48 (P1): Improper use of semicolon. Remove the semicolon L55 (P3): Commas (or parentheses) needed after EGC and frozen: "EGC, both liquid and frozen, varies". L67

(P3): Poor sentence structure. Suggested correction: "However, it is also possible that liquid". L112-113 (P5): The sentence starting with "During winter" has poor structure. L122-123 (P5): Replace "his" with "its". "way of data sampling" is poor word choice. L133 (P5): Comma required after "If requested". "current" should be plural. L135 (P5): "lesser" should be "less". L184 (P7): Replace "have been" with "were". "Large part" should say "A large part". L194 (P8, Figure 3 caption): Poor sentence structure. L204 (P10): Only SN 127 is equipped with an oxygen sensor, per table 2. Either it is missing in the table for SN 558, or this line should be modified, ". . .conductivity, pressure, oxygen (SN127 only) and optical parameters. . ." L226 (P10): The Section 3.2 header is confusing and should be revised. L239 (P11): "byt" should be "by". L371-373 (P20): This sentence is poorly worded and should be revised. Figure 5 (P21): I think there are too many colors used in this figure. I suggest eliminating black (use red and green solid and dotted for the upper plots), or use red and blue only in both subplots (including dotted in the upper plots). Table 4 (P24): Please double check your references to processing steps in column 1. Many of them reference the wrong processing step (ie B.5 instead of B.7 in the last row).

---

## Referee Comment (RC2) · Anonymous Referee #2 · 22 Jan 2019

In this manuscript the authors present temperature, salinity and drift data collected using autonomous Seagliders in two consecutive summers (2014 and 2015) near the ice edge of the East Greenland Current (EGC) and the western Greenland Sea in the upper 1000 meters. The effort was a part of the "Variation of freshwater on the western Nordic Seas"-project. The manuscript also contains a detailed presentation of the data processing steps and a brief description of the observations. The final data sets are easily accessible through the link provided. The EGC carries freshwater from the Arctic Ocean into the Nordic Seas and to the northern North Atlantic; regions with open ocean deep convection. High quality, multi-year data from this, rather inaccessible, region is of interest to the scientific community. However, I suggest the following revisions be

made.

Specific comments: Overall, the usefulness of the data set to the scientific community should be discussed to a much larger degree. Also, is there other data that this dataset can complement? Are there any references in connection to the project "Variation of freshwater on the western Nordic Seas"?

When describing the glider set up in 3.1 you could mention the pre-deployment tank tests and the sail specifications here.

Although the different steps in the data processing are thoroughly explained, I suggest looking over the structure of the presentation of the data processing and data quality (3.3 and 3.4). While I can understand the reason for structuring it this way, I found it made me go back and forth between these sections a lot trying to make sense of what happened when.

When reading about the individual corrections (below B.7 in section 3.3) it is not clear to me what this actually included (everything mentioned in the bullet points? or some?), there is some more information in 3.4.3 which might have been useful to know when reading the previous section, but it's still not very clear. What criteria were used to determine which data were erroneous in the different bullet points e.g. regarding outlier profiles, wrong values, large gaps etc.?

Row 334 - have these spikes been clearly flagged so that they can easily be taken into account when someone is using the data set? Or is it likely that they were removed in another processing step after A.6?

Rows 337-339 - so why was this method not used here?

I would suggest a paragraph at the end of section 3 where the authors summarize and discuss the quality of their processed data and how the data could be used (or should not be used).

Please add a conclusion at the end of section 4.

In the online data product: I suggest not changing the Operation number to NA in the "drift"-files. I understand that it is because there is no hydrography parameters available for that dive, however I suggest adding another column instead to flag this.

The number of dives for glider 127 in 2014 seems to be 258 in the online file, but is listed as having 220 dives in Table 1.

Technical comments: Row 55 should have commas around "both liquid and frozen"

Row 58 should be "low salinities were frequently observed"

Row 105 and 107 "is flowing" should probably rather be "flows"

Row 122 and 123 "his" should be "its"

Row 130 "support the realization"?

Row 135 too many "the"

Row 153-156 wrong line spacing

Row 184 "have been" should be "were"

Row 185 "already beginning of" should be "already in the beginning of"

Row 194 "the maps base on" should be "the maps are based on"

Row 208-209 – perhaps swap the two URLs to give better line break? Looks odd now

Row 213 and 214 "data of" should perhaps be "data from"?

Row 218 "track or" should be "track, or"

Row 226 Title needs rewording

Row 229 "follows basically" should be "basically follows"

Row 231 "miss-alignment" should be "misalignment"

[Figure]

Row 239 "byt" should be "by"

Row 257 "sampling rate information" should be "sampling rate, information"

Row 259 "cell but" should probably be "cell; instead" or similar

Row 281 "It was also analyzed if" should be "An analysis was also made to determine if"

Row 281 "show" should be "showed"

Row 282 "if they can be used both or not" should be something like "if both could be used or not", or perhaps "which of them, if any, could be used" or something similar.

Row 289 – "divice" should be "device"

Row 291 Insert blank line On page 14-15, the "individual steps of table 2" (there are no row numbers here) In B.4 "interested to analyze" should be "interested in analyzing" In B.5 "iterative" should be "iteratively"

In B.5 "This is other than" should be "This is different from" or "This works differently than"

In B.6 "Fortunately for none of the missions reported here, systematic differences between down and up-casts were visible." should be something like "Fortunately, no systematic differences between down and up-casts were visible for any of the missions reported here"

Row 322 "if conductivity laged temperature" should be "if conductivity lagged behind temperature"

Row 325 "not successful at whole" should be, depending on what the intended meaning is, be something like "not successful overall" or "not successful at all times" or possibly something else.

Row 333 "hereupon" should probably be "therefore"

Row 391 "the criteria of stable density was applied" should be "the criteria of stable density were applied"

Row 394 "exemplarily" means "In an exemplary manner; ideally, admirably" – so "again exemplarily for glider 127 during the mission 2015" should be changed to something like "where again glider 127 during the mission 2015 is used as an example"

Row 408 "and thus demonstrate" should be "and thus demonstrates"

Rows 445-462 – here you change between present and past tense back and forth several times, which is confusing. Pick one – preferably past tense – and apply it consistently to this section.

Row 469 – as on row 394, the word "exemplarily" can't be used like this - rephrase

Row 470 "In the right column map extracts" should be "In the right column, map extracts"

Row 471 "For 2014 also the ice edge at the arrival time of the glider at the edge is included in the map" should be "For 2014, the ice edge at the arrival time of the glider at the edge is also included in the map"

Rows 490, 510 and 526 "toke place" should be "took place"

Row 520 "upper 55m" should be "upper 55 m"

Row 537 "making public available the UAE toolbox" should be "making the UAE toolbox publicly available"

Row 542 "Harald Rohr,OPTIMARE" should be "Harald Rohr, OPTIMARE" (missing a space after the comma)

Row 545 "We like to thank" should be "We would like to thank"

---

## Author Comment (AC1) · 20 Mar 2019

SPECIFIC COMMENTS

L32 (P1): For consistency in the sentence that refers to figure 1, add "in West Spitsbergen Current (WSC)" after "along their eastern rim".
**For consistency in the sentence and for a complete description of the flow along the eastern rim, we add the West Spitsbergen Current as well as the Norwegian Atlantic Current; the Norwegian Atlantic Current is also introduced in Figure 1!**
**New Text:**
The Nordic Seas are shaped by a strong near-surface salinity contrast arising from the northward flow of saline Atlantic Water along their eastern rim in the Norwegian Atlantic Current (NwAC) and West Spitsbergen Current (WSC) and the southward flow of fresh Polar Water and sea ice along their western rim in the East Greenland Current (EGC) (Fig. 1).

Figure 1 caption: I suggest the acronym definitions get moved to the bottom of the figure caption, below the description of the red arrows. Additionally, L48 (P1): consider rewording this sentence. The gradient in the red arrows is what is indicative of the cooling, so this should be stated first. Suggested rewording: "The red to yellow arrows indicate the relative cooling of the warm, saline Atlantic Water as it flows through the Nordic Seas and Arctic Ocean."
**Changes are made as suggested.**
**New text and Figure:**

[Figure]

**Figure 1: The map shows the Nordic Seas. Topographic contours are given on the basis of RTOPO2 (Schaffer et al., 2016): the 1000 m contour is marked in red, the 3000 m and 300 m contours in black, and the 2000 m, 500 m and 200 m contours in gray. The inlet marked by the full blue line shows the area of Fig. 2, the inlet marked by the dashed blue line shows the area of Fig. 3.**
**Red to yellow arrows indicate the cooling of the warm and saline Atlantic Water as it flows through the Nordic Seas and Arctic Ocean. The blue arrows indicate the flow of cold and fresh Polar Water through the Nordic Seas.**
**EGC – East Greenland Current, WSC – West Spitsbergen Current, NwAC – Norwegian Atlantic Current, FS – Fram Strait, GS – Greenland Sea, NT – Norske Trough.**

L52-56 (P2-3): This paragraph should be more descriptive and/or reorganized. A few brief clarifying statements could help. The first sentence lacks temporal context (are the numbers listed a general average? Are the numbers listed percentages of total output over a year, during a specific season, or?). You then mention the temporal variability in freshwater transport in the EGC; how does this variability relate to the EGC's contribution to total freshwater output (the 50-75% you mention)? Additionally, variability in freshwater transport can be due to either variability in current speeds and/or variability in total volume fraction of freshwater to total seawater. Which (or both) is more influential here?
Rewritten.
**We focus on the expected long-term development of Arctic outflow of freshwater with the EGC and only mentioned variability and measurement uncertainties. This paragraph is now more detailed than the rest of the Introduction. We think that describing variability and uncertainties in even more detail is beyond the scope of the manuscript.**
**New text:**

Freshwater leaves the Arctic with the EGC in liquid form and as sea ice. For the liquid export de Steur et al. (2014) estimated 2100 km³/yr over the period 2000-2010. The Fram Strait export of freshwater in sea-ice, averaged over the winters 2003-2008, is estimated to have been 2100 km³/yr (Spreen et al., 2009). The annual average for 2000-2010 is 1900 km³/yr, when data gaps are filled using the average seasonal cycle (Hain et al., 2015). Haine et al. (2015) related these fluxes to other fluxes into and out of the Arctic as well as to the freshwater reservoir of the According to these considerations, liquid and sea ice fluxes with the EGC to the Nordic Seas account for almost 50 % of the total freshwater outflow from the Arctic for 2000 to 2010, with almost no changes in relation to the time span 1980 to 2000. Haine et al. (2015) expect an increase of the liquid outflow through Fram Strait by around 100 % for the next century, as at present the freshwater reservoir of the Arctic is increasing due to increasing river runoff and precipitation minus evaporation and due to ice melt. The sea ice outflow is expected to decrease due to the reduction of sea ice in the Arctic. This overall trend is anticipated to be superimposed by seasonal, interannual and decadal variability, mainly forced by variability in the wind-field (for a detailed discussion of wind-forced variability see Hain et al., 2015). Additional variability in the sea ice flux is introduced by the interplay of sea ice thickness, velocity and area (Smedsrud et al., 2011; Hansen et al., 2013; Spreen et al., 2009). Finally, large uncertainties are associated with this estimates, as the liquid freshwater flux, particularly the part close to the surface, as well as the different components of the sea ice flux are difficult to observe (Hansen et al., 2013; Spreen et al., 2009, de Steur et al., 2009; Hain et al. ,2015; and references included).

L140-142 (P5-6): The parenthetical description of "roll" does not help to clarify this sentence. "Turn to the left/right" could refer to roll or yaw. Either eliminate, or use the actual axis of rotation in your description.
**We clarified the section about steering of the gliders. The terms 'pitch' and 'roll' are terms of the glider manuals (but not 'yaw'). Therefore, I would like to use them here also.**
**New text:**

To control the roll of the instrument an additional weight is fixed axial asymmetric at the battery pack. As gliders behave like an "under-water sailplane", turning the battery to the right or left forces the glider to turn horizontally to the right or left accordingly.

Section 2.3 (P6): Table 1 is very thorough and contains a lot of good information. You should summarize more of the pertinent information in the text portion of this section. Consider moving the last paragraph of section 2.2 to section 2.3. Your first explanation of the reasoning behind the mission planning is too general (L146-147); include more specifics up front. L154: "but later concentrated on a southeast to northwest section" – why? Also, please explain the voltage-cutoff and unstable flight behavior during summer 2014 (glider 127). What were the causes? It appears these issues were resolved for the second deployment of glider 127 but this is not well explained.

**We included more information about the motivation for the mission design and about the course of the missions.**

**We observed different flight behavior between gliders and also of glider 127 between summer 2014 and 2015, but the explanation of these differences is difficult. Glider 127 is our oldest glider and we refused to upgrade important components of it during refurbishment. Instead we trust in our knowledge about malfunctioning and how to deal with it. The stability of the flight in a certain mission also depends on a smooth transportation from the refurbishment to the mission and the nuances of success of ballasting and trim during an individual refurbishment. But the description of these things is beyond the scope of the paper.**
**New text:**

During summers 2014 and 2015, Seaglider missions were carried out in the western GS. The goal was to capture the spreading of freshwater from the western rim into the inner Nordic Seas. For this goal we run the glider(s) along a section between the deep GS basin and the EGC. Repeating the section in 2015 allowed observation of the variability of the spreading both during the course of the individual summers as well as between the two summers.

In 2014, the measurements started with an east to west section. Because of the ice coverage in the early summer, the mission had to be changed later to a southeast to northwest section (Fig. 2) perpendicular to the isobaths. For comparability, the southeast to northwest section was carried out in 2015 too (from 75°N/2°W to 76°N/6°W in 2014 and to 76°30'N/7°20'W in 2015). Only the last section conducted with glider 127 in 2015 was displaced to the north to capture also the Norske Trough. Table 1 summarizes information about both missions.

Section 2.4 (P7): Your description of the differences in ice regimes between the two seasons is well detailed. What are the implications to the datasets in terms of their capabilities, comparability with each other and with other datasets, etc?
**We added a few sentences to give a perspective, but we consider it is beyond the scope of this manuscript to analyze this in detail.**
**New text:**
The glider missions in summer 2014 and summer 2015, give insight into the distributions of temperature and salinity in the upper part of the water column. In summer 2015, the distribution was also observed in regions, where the ice coverage just disappeared. The observations can be interpreted in relation to the different ice coverage (see section 4).

Figure 3 (P8-9): On P9 why is one of the glider tracks represented by a dotted line but this is not described in the figure caption?
**This is the track of 558 at the end of the mission to the recovery position, where still measurements were taken. We added a description in the figure caption.**
**New text:**
**For each year, a sketch of the respective glider sections is added to the map (red lines and blue dots; the red dashed line in the bottom left map shows the track of glider 558 to the recovery position).**

L228 (P10): Explain how the glider data are comparable to CTD data (in sample frequency,

resolution?).
**We rewrote the beginning of the section.**
**Details about frequency and vertical resolution of the glider measurements compared to ship CTD**
**measurements were given below in the list (now L298-315)**
**New text:**
A glider measures temperature, conductivity and pressure while it is moving vertically and horizontally through the water. The relation between horizontal and vertical movement during our missions was 2:1 and an approximate localization of each measurement is possible with the start and end position of each dive. During the processing, the data were handled like ship-based CTD measurements consisting of temperature and conductivity reading related to ideally monotonously increasing pressure readings. Thus, the processing for these data basically follows the processing for ship-based CTD data.

L280-281 (P12): Describe your matchup criteria (spatial and temporal) quantitatively
(what does "close" mean?). I see the spatial criteria is listed in B.7 but should be
mentioned here (or B.7 referenced).
**We did not implemented this comment, as we don't want to mix up the different sub-sections.**
**The idea was to have first (3.2) the motivation for our effort, second (3.3) a preferably general**
**description of the different steps of the data processing and last (3.4) specific details concerning**
**the data sets processed here, concerning problems faced during the processing and decisions**
**made to solve the problems.**

B.4 (P15): Is the variable 'numrec' the same as 'NOBS[#]'? The latter is what is used
in the dataset available on PANGEA. This final column in the published dataset has
missing values for select entries (ie PS93_SG127_hydrography profile 348, 8dbar, Direction
1). What does this mean? How can there be < 1 obs used for a line entry, when
all other fields are populated?
**We changed the name to NOBS. I did not realize that the names were changed in the final data set at**
**PANGAEA. An explanation for empty NOBS is added.**
**New text:**
In the final data set the variable NOBS gives the number of observations from which 2 dbar-means were

calculated. If NOBS is empty for a certain line of data, values for temperature, conductivity, salinity and density

were interpolated.

B.5 (P16): The steps outlined here seem circular. Why not smooth conductivity? Do
you have a reference for this method and can you describe why you chose the thresholds
that you did? What is the mean difference between original and recomputed
values?
**This way of smoothing was motivated by the fact that we expect to have a stable stratification away**
**from mixing zones.**
**New text:**
Particularly there were small instabilities in the density stratification, which we considered not to be real.

L333-335 (P18): Why did you choose not to include quality flags in your dataset, especially
since you chose to leave uncorrected spikes? It seems this should be indicated
in the dataset for the user in some way, either as a header note or (preferably) quality
flags.
**Although quality flag standards are developed comparable to Argo standards within the EGO**
**community (https://www.ego-network.org/), no standard data processing and quality control is**
**established yet. Thus, setting flags would be subjective. The changes made in B.4 allow**
**identification of interpolated values. Additionally, we incorporated an Annex with a list of the**
**profiles with spikes in the thermo/halocline.**
**New Text:**
In the final data set the variable NOBS gives the number of observations from which 2 dbar-means were

calculated. If NOBS is empty for a certain line of data, values for temperature, conductivity, salinity and density

were interpolated.

Annex:
List of individual profiles with spikes in the thermo/halocline.
For details see Section 3.4.3.

Glider 127 2014:
Dive no: 10-13, 17, 11, 24, 76, 82, 206-208, 212-214, 220-227, 229-231, 233-234

Glider 558 2014:
Dive-no: 1, 3-13, 15-25, 85-86, 91-93, 101-103, 110-112, 116-121, 125-127, 390

Glider 127 2015:
Dive-no: 2-7, 9-17, 19-32, 34-67, 75-77, 106-107, 109-115, 117-124, 167-226, 230, 233, 329-420.

The dive-no is named *observation number* in PANGAEA.

L395 (P21) and Figure 6: Your final statement of a reduction in salinity variability of
50% is too vague. It really only applies to Figure 6, glider 127, mission 2015 in the
deep layer (which is difficult to see in the plot). Your plot should better exemplify this
(zoom in on deep layer instead of surface?), and text should be more descriptive (is it
an average reduction, and what is the std of the difference in variability?). Similarly, in
table 4 'variability reduction of salinity' should better describe the where the numbers
came from, I assume they are averages of the differences in variability at each depth
interval? I see an average reduction of mean variability of between _8-10%, _30-60%,
and _13-48% in the three layers, respectively, and across all missions.
**We rewrote part of Section3.4.4.2 and added some comments and numbers in Table 4.**
**We think it will not help to clarify this paragraph, if we additionally introduce the std of the**
**variability. A new version of Figure 6 is included with extracts of 250 to 1000 dbar.**
**New text:**
To quantify the noise reduction resulting from step B.5, where the criterion of stable density was applied, we

calculated the variability of a profile before and after the step. The variability is defined here as the difference

between consecutive values of salinity in a profile. Figure 6 shows the variability for all individual salinity

profiles; again, glider 127 during the mission 2015 was used as an example. In the upper 80 dbar the noise

reduction is of order 10 %, but between 45% and 63% below. The average reduction for the whole depth range is

22 % for glider 127 in 2014, 13 % for glider 558 in 2014 and 33 % for glider 127 in 2015.

[Figure]

[Figure]

**Figure 6: (Top) Variability for all individual salinity profiles before (red) and after (blue) smoothing of density, (left) for the total profile depth, (right) for the upper 200 dbar; again, glider 127, mission 2015 was used as an example. (Bottom) an individual salinity (left) and density (right) profile before (red) and after (blue) smoothing of density. Extracts of 250 to 1000 dbar are inserted.**
**New part of Table 4:**

| Average variability* of salinity in the surface layer, below, and in the whole depth range before step B.5 → after step B.5, and variability reduction in percentage.

(step B.5,
see Section 3.4.4.2)
*variability is defined as the difference between consecutive values of salinity in a profile (see Section 3.4.4.2) | 0-80 dbar
0.051 → 0.046
→ 10 %
80-1000 dbar
0.0019 →0.0007
→ 63 %
0-1000 dbar
0.0058 → 0.0045
→ 22 % | 0-80 bar
0.073→ 0.067
→ 8 %
80-1000 dbar
0.0011 → 0.0006
→ 45 %
0-1000 dbar
0.0069 → 0.0060
→ 13% | 0-80 dbar
0.028 → 0.025
→ 11 %
80-1000 dbar
0.0025 → 0.0013
→ 48 %
0-1000 dbar
0.0046 → 0.0031
→ 33 % |
|---|---|---|---|

L407-409 (P22): Great.

Table 4 (P24): Given the data quality and accuracies you outline in table 4, what are the limitations (are there any?) to use of the dataset, in terms of better understanding fluctuations/dynamics of changing freshwater fluxes in Nordic Seas?
Section 4: Interesting observations stemming from this dataset; the multi-year span provides great context for comparisons. This section lacks any concluding remarks.
**At the end of section 4 conclusive remarks are added, which also relate to section 3.4.**
**New text:**
The presented distributions of temperature and salinity, measured along sections from the inner GS to the EGC during summer 2014 and summer 2015, show signs of freshwater intrusions close to the surface. The development within a single summer as well as the interannual differences are demonstrated. The freshwater intrusions are not masked by the inaccuracies of the measurements, as we described in detail in Section 3, as the absolute difference between the Polar Surface Water and the Arctic Intermediate Waters is of order 4-6 K for

temperature and 2-4 for salinity. For further analyses, one has to take into account that in opposite to ship-based CTD sections, glider sections are never "quasi-synoptic". Thus, the combination of low time resolution and high spatial resolution provided by glider measurements must be considered, when deriving quantitative conclusions from the observed distributions.

TECHNICAL CORRECTIONS

L48 (P1): Improper use of semicolon. Remove the semicolon
**Done**

L55 (P3): Commas (or
parentheses) needed after EGC and frozen: "EGC, both liquid and frozen, varies".
**Done**

L67 (P3): Poor sentence structure. Suggested correction: "However, it is also possible that liquid".
**Done**

L112-113 (P5): The sentence starting with "During winter" has poor structure.
**Changed.**

L122-123 (P5): Replace "his" with "its". "way of data sampling" is poor word choice.
**Changed. "Its data sampling scheme" instead of "way of data sampling"**

L133 (P5): Comma required after "If requested". "current" should be plural.
**Done**

L135 (P5): "lesser" should be "less". L184 (P7): Replace "have been" with "were". "Large part" should say "A large part".
**Done**

L194 (P8, Figure 3 caption): Poor sentence structure.
**Rewritten.**
**New text:**
**Figure 3: The development of the ice cover in the western Greenland Sea during the time span of the glider missions in summer 2014 and summer 2015.**
**Left column for summer 2014, right column for summer 2015. Month and day of the individual ice concentration maps are given in the upper left corner. The maps are based on ice concentration data made available by DRIFT&NOISE (driftnoise.com). For each year, a sketch of the respective glider sections is added to the map (red lines and blue dots; the red dashed line in the bottom left map shows the track of glider 558 to the recovery position). Black contours give the 3000 m, 1000 m and 300 m depth contour based on RTOPO2 (Schaffer et al., 2016). The location of the map is shown as inlet in Fig. 1 with a blue dashed line.**

L204 (P10): Only SN 127 is equipped with an oxygen sensor, per table 2. Either it is missing in the table for SN 558, or this line should be modified, ". . .conductivity, pressure, oxygen (SN127 only) and optical parameters. . ."
**Done**

L226 (P10): The Section 3.2 header is confusing and should be revised.
**Changed to:**
**3.2 Glider data processing**

L239 (P11): "byt" should be "by".
**Done**

L371-373 (P20): This sentence is poorly worded and should be revised.

**Done**

Figure 5 (P21): I think there are too many colors used in this figure. I suggest eliminating black (use red and green solid and dotted for the upper plots), or use red and blue only in both subplots (including dotted in the upper plots).
**We would like to leave the line colors/styles in Figure 5 as it is, because using only red and green in the top figures doesn´t allow to separate the lines between down- and up-cast.**
**We choose different colors for the bottom figures to realize that not temperature and conductivity but a different quantity is shown there. We could replace red in the bottom figure by another color but red, green, blue and black are the most visible ones.**

[Figure]

Table 4 (P24):

Please double check your references to processing steps in column 1. Many of them reference the wrong processing step (ie B.5 instead of B.7 in the last row).
**Done, changed.**

---

## Author Comment (AC2) · 20 Mar 2019

Specific comments: Overall, the usefulness of the data set to the scientific community should be discussed to a much larger degree. Also, is there other data that this dataset can complement? Are there any references in connection to the project "Variation of freshwater on the western Nordic Seas"?
**We are only aware of SST and SSS data, but there have been no hydrographic cruises during the glider missions in the same region.**
**Our topic was the only component in the DFG research group, which worked in the Nordic Seas.**

When describing the glider set up in 3.1 you could mention the pre-deployment tank tests and the sail specifications here.
**Done**
**New text:**
Temperature and conductivity sensors have been calibrated by Sea-Bird (www.seabird.com) and the instruments were refurbished before the missions. The refurbishment included trimming and ballasting with tank tests and sea-trials.

Although the different steps in the data processing are thoroughly explained, I suggest looking over the structure of the presentation of the data processing and data quality (3.3 and 3.4). While I can understand the reason for structuring it this way, I found it made me go back and forth between these sections a lot trying to make sense of what happened when.
**We leave Section 3.2 to 3.4 as it was.**
**The idea was to have first (3.2) the motivation for our effort, second (3.3) a preferably general description of the different steps of the data processing and last (3.4) specific details concerning the data sets processed here, concerning problems faced during the processing and decisions made to solve the problems.**
**We got the feeling that this structure is more clear than putting all information about each step of the processing in one single paragraph.**

When reading about the individual corrections (below B.7 in section 3.3) it is not clear to me what this actually included (everything mentioned in the bullet points? or some?), there is some more information in 3.4.3 which might have been useful to know when reading the previous section, but it's still not very clear. What criteria were used to determine which data were erroneous in the different bullet points e.g. regarding outlier profiles, wrong values, large gaps etc.?
**We moved information from below B.7 (Section 3.3) to 3.4.3 and added information to clarify the decisions made. We did this to have as little redundant information as possible and follow our idea of the structure as described above.**
**New text:**
**3.4.3 Visual inspection of the temperature, conductivity, salinity, density and vertical velocity profiles**

(data processing step *individual corrections*)

By visual inspection of all individual profiles at different steps of the processing, several individual faulty values

or profiles are detected:

- Spikes in salinity in the depth range of the thermo/halocline. These were removed, if they exceeded 0.1 (see Section 3.4.2)

- Wrong values during the apogee, which were not removed by the criterion w < 5 cm/s. These show up as temperature and conductivity values, which are far apart from the continuous profile, although the pressure did not change; they were removed.

- Outlier profiles of conductivity. Profiles, which are considerably separated from the entity of profiles of a mission, were removed.

- Profiles with large gaps in the depth of the largest gradient. If the gaps exceeded a depth range larger than the typical depth range of the thermo /halocline (> 10 dbar) the profiles were removed

- Incomplete profiles. When the dive was aborted by the glider-intrinsic software after an uncommanded change in the bleed counts of the vertical buoyancy device, these profiles were removed.

No individual temperature, conductivity or salinity values were removed, but always complete data lines or even the whole profiles were removed before the interpolation to 2 dbar levels took place. This results in a reduction of the original data sets between 2 % and 5% (Table 4).

Row 334 - have these spikes been clearly flagged so that they can easily be taken into account when someone is using the data set? Or is it likely that they were removed in another processing step after A.6?
**We explained our motivation for not removing these profiles with one sentence..**
**The reasons for not flagging them are described in the reply to reviewer #1:**
**Although quality flag standards are developed comparable to Argo standards within the EGO community (https://www.ego-network.org/), no standard data processing and quality control is established yet. Thus, setting flags would be subjective. The changes made in B.4 allow identification of interpolated values. Additionally, we incorporated an Annex with a list of the profiles with spikes in the thermo/halocline.**
**New Text:**
We decided to leave the decision how to deal with the spikes to the users of the data set. To help identification of affected profiles we list them in the Annex. The spikes will possibly level out during gridding or averaging routines in further processing. For example, Queste et al. (2016) developed a method to deal with glider measurements across sharp gradients. They built composite profiles from the downcasts between the surface and the thermo-/halocline and from the upcasts between maximum depth and thermo-/halocline and combined these in a gridded data set.

**B.4**
**..**
In the final data set the variable NOBS gives the number of observations from which 2 dbar-means were calculated. If NOBS is empty for a certain line of data, values for temperature, conductivity, salinity and density were interpolated.

Annex:
List of individual profiles with spikes in the thermo/halocline.
For details see Section 3.4.3.

Glider 127 2014:
Dive no: 10-13, 17, 11, 24, 76, 82, 206-208, 212-214, 220-227, 229-231, 233-234

Glider 558 2014:
Dive-no: 1, 3-13, 15-25, 85-86, 91-93, 101-103, 110-112, 116-121, 125-127, 390

Glider 127 2015:
Dive-no: 2-7, 9-17, 19-32, 34-67, 75-77, 106-107, 109-115, 117-124, 167-226, 230, 233, 329-420.

The dive-no is named *observation number* in PANGAEA.

Rows 337-339 - so why was this method not used here? Because the decision, how to deal with these profiles, depends on the specific interest of the users.
**We added an explanation. See last reply.**

I would suggest a paragraph at the end of section 3 where the authors summarize and discuss the quality of their processed data and how the data could be used (or should not be used).
Please add a conclusion at the end of section 4.
**At the end of section 4 some conclusions are added, which also relate to section 3.4.**
**New Text:**
The presented distributions of temperature and salinity, measured along sections from the inner GS to the EGC during summer 2014 and summer 2015, show signs of freshwater intrusions close to the surface. The development within a single summer as well as the interannual differences are demonstrated. The freshwater intrusions are not masked by the inaccuracies of the measurements, as we described in detail in Section 3, as the absolute difference between the Polar Surface Water and the Arctic Intermediate Waters is of order 4-6 K for temperature and 2-4 for salinity. For further analyses, one has to take into account that in opposite to ship-based CTD sections, glider sections are never "quasi-synoptic". Thus, the combination of low time resolution and high spatial resolution provided by glider measurements must be considered, when deriving quantitative conclusions from the observed distributions.

In the online data product: I suggest not changing the Operation number to NA in the "drift"-files. I understand that it is because there is no hydrography parameters available for that dive, however I suggest adding another column instead to flag this.
**PANGAEA was requested to change it that every observation has an operation number.**

The number of dives for glider 127 in 2014 seems to be 258 in the online file, but is listed as having 220 dives in Table 1.
**For "dive" 221 to 258 only position and drift data available. We add this information to table 1.**
**Part of table 1:**

|  | 2014/08/21 voltage-cutoff; surface drift until recovery; position and drift data for "dive" 221-258 are available |  |  |
|---|---|---|---|

Technical comments:
Row 55 should have commas around "both liquid and frozen"
**Done**

Row 58 should be "low salinities were frequently observed"
**Done**

Row 105 and 107 "is flowing" should probably rather be "flows"
**Done**

Row 122 and 123 "his" should be "its"
**Done**

Row 130 "support the realization"?

**Changed to "support the steering of the glider'**

Row 135 too many "the"?
**one "the" deleted**

Row 153-156 wrong line spacing
**Changed**

Row 184 "have been" should be "were"
**Changed**

Row 185 "already beginning of" should be "already in the beginning of"
**Done**

Row 194 "the maps base on" should be "the maps are based on"
**Done**

Row 208-209 – perhaps swap the two URLs to give better line break? Looks odd now
**Done**

Row 213 and 214 "data of" should perhaps be "data from"?
**Done**

Row 218 "track or" should be "track, or"
**Done**

Row 226 Title needs rewording
**Changed to" 3.2 Glider data processing"**

Row 229 "follows basically" should be "basically follows"
**Changed**

Row 231 "miss-alignment" should be "misalignment"
**Done**

Row 239 "byt" should be "by"
**Done**

Row 257 "sampling rate information" should be "sampling rate, information"
**Done**

Row 259 "cell but" should probably be "cell; instead" or similar
**Done**

Row 281 "It was also analyzed if" should be "An analysis was also made to determine if"
**Done**

Row 281 "show" should be "showed"
**Done**

Row 282 "if they can be used both or not" should be something like "if both could be used or not", or perhaps "which of them, if any, could be used" or something similar.
**Done**

Row 289 – "divice" should be "device"
**Done**

Row 291 Insert blank line On page 14-15, the "individual steps of table 2"
**Done**

(there are no row numbers here)
**because it is formatted as a table**

In B.4 "interested to analyze" should be "interested in analyzing"

**Done**

In B.5 "iterative" should be "iteratively"
**Done**

In B.5 "This is other than" should be "This is different from" or "This works differently than"
**Done**

In B.6 "Fortunately for none of the missions reported here, systematic differences between down and up-casts were visible." should be something like "Fortunately, no systematic differences between down and up-casts were visible for any of the missions reported here"
**Done**

Row 322 "if conductivity laged temperature" should be "if conductivity lagged behind temperature"
**Done**

Row 325 "not successful at whole" should be, depending on what the intended meaning is, be something like "not successful overall" or "not successful at all times" or possibly something else.
**Done**

Row 333 "hereupon" should probably be "therefore"
**Done**

Row 391 "the criteria of stable density was applied" should be "the criteria of stable density were applied"
**Changed to criterion**

Row 394 "exemplarily" means "In an exemplary manner; ideally, admirably" – so "again exemplarily for glider 127 during the mission 2015" should be changed to something like "where again glider 127 during the mission 2015 is used as an example"
**Changed**

Row 408 "and thus demonstrate" should be "and thus demonstrates"
**Done**

Rows 445-462 – here you change between present and past tense back and forth several times, which is confusing. Pick one – preferably past tense – and apply it consistently to this section.
**We switched to past tense. But there are some statements that should be present.**

Row 469 – as on row 394, the word "exemplarily" can't be used like this – rephrase
**Done**

Row 470 "In the right column map extracts" should be "In the right column, map extracts"
**Done**

Row 471 "For 2014 also the ice edge at the arrival time of the glider at the edge is included in the map" should be "For 2014, the ice edge at the arrival time of the glider at the edge is also included in the map"
**Done**

Rows 490, 510 and 526 "toke place" should be "took place"
**Done**

Row 520 "upper 55m" should be "upper 55 m"
**Done**

Row 537 "making public available the UAE toolbox" should be "making the UAE toolbox publicly available"
**Done**

Row 542 "Harald Rohr,OPTIMARE" should be "Harald Rohr, OPTIMARE" (missing a space after the comma)
**Done**

Row 545 "We like to thank" should be "We would like to thank"
**Done**

---

## Author Response (AR2)

**Topical Editor Decision: Publish subject to minor revisions (review by editor)** (02 May 2019) by Robert Key
Comments to the Author:
Minor technical details
In a few places Haine is spelled "Hain"
Base station is spelled as one word. This is acceptable as part of a product name, but should be separated otherwise.

Comment:
I agree with both reviewers that the data set would have been better if the data had been flagged (spikes, etc.). By not doing so you necessitate that every user go through that procedure of treat the values as real. Even though your decisions might be subjective, no one else is likely to be as competent as you to do this task. The end result is that far fewer people will use the data and/or the results will be compromised by this choice. Nevertheless, I believe that this decision is up to the data producers and therefore recommend the manuscript/data be published as is.

06 May 2019

Dear Robert,

Thank you very much for intensive reading and tracking the last misspellings. I corrected all Hain to Haine and changed basestation to base station if it is not referred to the KONGBERG basestation.
And thank you very much for your understanding that we did not add quality flags.

The changes to the PANGAEA data set, announced in the response to the reviewers, have been submitted to PANGAEA. But until now we have no answer, when they will be incorporated. They will be described in the ERRATUM. The ASCII files will be supplemented by netcdf files.

Hopefully everything is in best order now,
Kind regards,
Katrin

[revised manuscript text omitted]

| | | |
|---|---|---|
| Section 1
Dives 1-55
Profiles 1-84
07/01-07/11 <----
Ice edge from 07/11 | 127 2014 section 1 | 127 2014 section 1 |
| Section 2 *
Dives 60-101
Profiles 93-147
07/11-07/18 ---->
Ice edge from 07/11 | 127 2014 section 2 | 127 2014 section 2 |
| Section 3
Dives 101-168
Profiles 148-259
07/18-08/02 <----
Ice edge from 08/05 | 127 2014 section 3 | 127 2014 section 3 |
| Section 4
Dives 168-220
Profiles 260-353
08/02-08/14 ---->
Ice edge from 08/05 | 127 2014 section 4 | 127 2014 section 4 |
| Section 5
Dives 221-258
Profiles 359-418
08/14-08/21 <----
Ice edge from 08/19 | 127 2014 section 5 | 127 2014 section 5 |

[revised manuscript text omitted]

75°00' N, 0°00' E/W
with RV Polarstern (PS85) | 2014/07/01
75°00' N, 0°01' W
with RV Polarstern (PS85) | 2015/07/14
75°45' N, 3°08' W
with RV Polarstern (PS93.1) |
| recovery | 2014/09/13
74°31' N, 1°58' W
with MV Plancius * | 2014/09/04
73°45' N, 16°16' W
with MV Ortelius * | 2015/10/05
75°45' N, 3°08' W
with RV Polarstern (PS94) |
| E-W sections | dive 1-101 | dive 1-120 | |
| SE-NW sections | dive 101-220 | dive 120-358 | dive 1-420
northern sect: dive 282-343
shelf-dive 128-151, 257-270,
shelf-dive with altimeter bottom
tracking: 292-316 |
| | | to recovery position (NE to SW)
dive 359-484 | |
| | 2014/08/21 voltage-cutoff;
surface drift until recovery;
position and drift data for "dive"
221-258 are available | | |
| total | 220 dives
52 days
179 dives to 1000 m
41 dives   500 m
typical distance for
500/1000 m dives: 2.8/3.8 km
total mission distance: 910 km | 484 dives
65 days
142 dives to 1000 m
342 dives   500 m
typical distance for
500/1000 m dives: 2.1/4.0 km
total mission distance: 1266 km | 420 dives
84 days
500 m   329 dives   1000 m
91 dives   500 m
typical distance for
500/1000 m dives: 1.7/4.8 km
total mission distance: 1678 km |

* We gratefully acknowledge the support by OCEANWIDE EXPEDITIONS.

[revised manuscript text omitted]

---

## Author Response (AR3)

20 May 2019

Dear Madame or Sir,

The upload for typesetting and publication includes 2 zip-archives with figures; one with all figures included in the manuscript in jpg, and one with all figures as they are shown in the manuscript. Only figure 1 appears at its own, all others are put together in a word-table including heading and comments. I extracted these tables for each numbered figure from the word document and transferred it to a pdf. I do not know how you will treat it.

Please keep me informed if I have to upload it differently.

Kind regards,
Katrin Latarius